# Simulation code for estimating external gamma-ray doses from a radioactive plume and contaminated ground using a local-scale atmospheric dispersion model

**Daiki Satoh**[1]*, **Hiromasa Nakayama**[1], **Takuya Furuta**[1], **Tamotsu Yoshihiro**[2], **Kensaku Sakamoto**[3]

1 Nuclear Science and Engineering Center, Japan Atomic Energy Agency, Naka-gun, Ibaraki, Japan,
2 Technical Services, East Technical Consultant, Hewlett-Packard Japan, Ltd., Koto-ku, Tokyo, Japan,
3 Center for Computational Science and E-system, Japan Atomic Energy Agency, Naka-gun, Ibaraki, Japan

* satoh.daiki@jaea.go.jp

**Data Availability Statement:** All relevant data are within the manuscript and its Supporting Information files.

## Abstract

In this study, we developed a simulation code powered by lattice dose-response functions (hereinafter SIBYL), which helps in the quick and accurate estimation of external gamma-ray doses emitted from a radioactive plume and contaminated ground. SIBYL couples with atmospheric dispersion models and calculates gamma-ray dose distributions inside a target area based on a map of activity concentrations using pre-evaluated dose-response functions. Moreover, SIBYL considers radiation shielding due to obstructions such as buildings. To examine the reliability of SIBYL, we investigated five typical cases for steady-state and unsteady-state plume dispersions by coupling the Gaussian plume model and the local-scale high-resolution atmospheric dispersion model using large eddy simulation. The results of this coupled model were compared with those of full Monte Carlo simulations using the particle and heavy-ion transport code system (PHITS). The dose-distribution maps calculated using SIBYL differed by up to 10% from those calculated using PHITS in most target locations. The exceptions were locations far from the radioactive contamination and those behind the intricate structures of building arrays. In addition, SIBYL's computation time using 96 parallel processing elements was several tens of minutes even for the most computationally expensive tasks of this study. The computation using SIBYL was approximately 100 times faster than the same calculation using PHITS under the same computation conditions. From the results of the case studies, we concluded that SIBYL can estimate a ground-level dose-distribution map within one hour with accuracy that is comparable to that of the full Monte Carlo simulation.

## Introduction

Radioactive materials can be dispersed in the environment due to nuclear power plant accidents and radiological terrorist acts using dirty bombs. Additionally, small amounts of

**Funding:** This study was partially supported by the Japan Society for the Promotion of Science (https://www.jsps.go.jp/english/index.html) KAKENHI Grant Number JP17K07017. In addition, one of the authors (TY) is affiliated with Hewlett-Packard Japan, Ltd. (HPE, https://www.hpe.com), which is a commercial company, and HPE provided support in the form of salaries for the author. The specific roles of the author are articulated in the "author contributions" section. The funders had no role in study design, data collection, and analysis, decision to publish, or preparation of the manuscript.

**Competing interests:** The authors have declared that no competing interests exist. One of the authors (TY) is affiliated with Hewlett-Packard Japan, Ltd. This does not alter our adherence to PLOS ONE policies on sharing data and materials.

radioactive materials are regularly released into the atmosphere from ventilation shafts of fuel reprocessing facilities. In such cases, external gamma-ray irradiation becomes a significant radiation exposure pathway. Thus, estimating the external gamma-ray doses at the ground level is indispensable for assessing possible public health risks. Radiation monitors can measure gamma-ray doses at their installation locations close to nuclear facilities; however, they cannot obtain complete spatiotemporal information in locations, especially urban areas, where radioactive materials have been intentionally or accidentally distributed. Hence, a simulation code to estimate external gamma-ray doses at any ground-level receptor is required to compensate for the lack of measured data and support the decision-making processes of early responders.

Prior research incorporated computational fluid dynamics (CFD) models for radioactive material dispersion with external gamma-ray dose models based on a point-kernel method with buildup factors [1–5]. These studies successfully evaluated ground-level gamma-ray dose rates based on cloud shine from overhead radioactive plumes; however, they were not designed to assess ground-shine doses from contaminants that were non-uniformly deposited on the ground after the plume passed. Recently, Zhang et al. [6] proposed a new point-kernel-based scheme for calculating ground shine. However, this model has not been incorporated into atmospheric dispersion models, and thus is not applicable to gamma-ray dose rate analysis along a timeline from the release of a radioactive plume to deposition on the ground. Moreover, the existing models cannot consider dose attenuation resulting from obstructions, such as buildings, which is crucial for simulating dose distributions in urban areas.

Recently, the Japan Atomic Energy Agency (JAEA) developed a new CFD code known as the local-scale high-resolution atmospheric dispersion model using large-eddy simulation (LOHDIM-LES) [7–11]. LOHDIM-LES can precisely predict a three-dimensional distribution of a radioactive plume in the air and surface contamination on the ground by simulating complex turbulent flows and dispersion behaviors based on large-eddy simulation (LES). However, this model cannot estimate external gamma-ray dose rates from exposure to the plume and the contaminated ground.

General purpose radiation-transport codes based on the Monte Carlo method, such as the particle and heavy-ion transport code system (PHITS) [12] developed by JAEA, are available for dose estimations by simulating radiation behavior in three-dimensional geometry. Although PHITS is a well-examined and reliable code for various radiations in a wide energy range [13], the full Monte Carlo simulation is computationally expensive with regard to computation time and machine resources. Hence, it is difficult to apply the PHITS code to dose estimations with a time constraint of a few hours [14], which is required during the initial response to a nuclear emergency.

With this background in mind, we developed a simulation code powered by lattice dose-response functions (which we called SIBYL) for estimating external gamma-ray doses from both an overhead radioactive plume and surface contamination on the ground. We designed SIBYL to combine with LOHDIM-LES and execute a sequential simulation from an atmospheric dispersion to a dose estimation. In addition, SIBYL can handle dose attenuation due to obstacles inside a simulation geometry and consider ground-elevation data. To perform reliable dose simulations with a computation time of less than one hour, SIBYL employed dose-response functions, which are matrices of pre-calculated doses per unit concentration, and a parallel computing algorithm. The results of SIBYL were compared with those of PHITS for typical cases to examine the validity of the SIBYL results. Furthermore, the performance of SIBYL's parallel computation was tested using a computer cluster system.

## Materials and methods

SIBYL calculates external gamma-ray doses from a radioactive plume and contaminated ground based on the results of atmospheric dispersion simulations obtained by LOHDIM-LES. Additionally, the dose-response functions that were used as an engine for dose calculation by SIBYL were prepared in advance with Monte Carlo simulations using the radiation-transport code PHITS. In this section, we briefly introduce LOHDIM-LES and PHITS, and then describe the computational procedure of SIBYL for calculating external gamma-ray doses. The set of cases investigated in this study is also presented here.

### Description of LOHDIM-LES

LOHDIM-LES [7–11] is an LES-based CFD code that employs the Eulerian approach in three-dimensional space with a Cartesian grid system. The code simulates turbulent flows and plume dispersions of radioactive materials over complex terrains containing buildings by numerically solving the governing equations, namely, the Navier–Stokes equation, the filtered continuity equation, and the scalar conservation equation. The building effect was considered as an external force in the Navier–Stokes equation. Previous studies [7–11] validated the reliability of LOHDIM-LES by comparing it with the results of wind tunnel experiments [15, 16] and field experiments [17] in an actual urban area under real meteorological conditions.

In this study, LOHDIM-LES read the data of the position and rate of the radioactive material emission, the wind field, the ground surface geometry (represented using the terrain data on topography), and the obstacles inside the target domain. Then the code generated the output data of the activity concentrations at the cells of three-dimensional and two-dimensional grids in the air and on the ground, respectively, along a time axis with a specified time step.

### Description of PHITS

PHITS [12] is a multi-purpose Monte Carlo code that simulates the transport and interaction of hadrons, leptons, and heavy ions with energies up to 1 TeV (per nucleon for ions) in arbitrary three-dimensional geometries using various nuclear reaction models and data libraries. In this study, we used the algorithm and database of the electron and gamma shower version 5 (EGS5) code [18] incorporated into PHITS version 3.02 to perform transport simulations of gamma rays emitted from radionuclides in the contaminant and secondary electrons generated by gamma-ray interactions with materials in the environment. The verification and validation of the EGS5 algorithm in PHITS for gamma-ray and electron transport were comprehensively tested with respect to various physical quantities in [13], and the calculation results exhibited sufficient consistency with the benchmark data.

For the calculation accuracy, we considered that PHITS could provide reference data for examining SIBYL validity. An in-house Fortran code was written exclusively for converting LOHDIM-LES output data into PHITS input data for dose calculations. Using this conversion code, the full Monte Carlo simulation by PHITS could follow the dispersion simulation by LOHDIM-LES. The results of PHITS shown in the results and discussion section were calculated using this procedure. Ground surface and obstacle geometries used in the LOHDIM-LES simulation were reproduced using the repeated-structures capability of the hexahedral lattice elements equipped in PHITS. The distribution of the radioactivity concentration outputted by LOHDIM-LES on a grid system was compiled with radiation-source data for PHITS using the same coordinate system as that of LOHDIM-LES. In addition, even though the simulation is computationally time- and resource-consuming, the coupling LOHDIM-LES and PHITS with the conversion code makes detailed retrospective dose assessments possible after the end of emergency-response phases.

## Computational procedure of SIBYL

SIBYL is a Fortran code to calculate ground-level external gamma-ray dose rates based on the activity concentrations of radionuclides in the plume and on the ground simulated by LOH-DIM-LES. A schematic of SIBYL's computational flow is shown in **Fig 1**. SIBYL constructs the simulation geometries (i.e., the ground surface including elevation and obstacle data), which are equivalent to those of LOHDIM-LES on a Cartesian grid system with the same grid resolution. Although the resolution can be changed in SIBYL by averaging or dividing the cells of LOHDIM-LES, the minimum resolution is set to 1 m. Radionuclide activity concentrations in the atmosphere and on the surface of ground are assigned to corresponding cells on the grid in units of $Bq/m^3$ and $Bq/m^2$, respectively. The activity concentrations expressed in $Bq/m^2$ for radionuclides on building walls are assigned to the adjacent cell on the grid in the atmosphere by converting the units to $Bq/m^3$ along with the conservation of total activity. SIBYL calculates the distribution of the ambient dose equivalent rate $\dot{H}^*(10)$ and the air kerma free-in-air rate $\dot{K}_{Air}$ at 1 m above ground inside the target area using the dose-response functions and activity concentrations. The size of the target area for the dose calculation is given in the SIBYL input. The treatments of the dose-response functions, obstacles, and elevations of the target cells in the dose calculations and the parallel computation of SIBYL are described in the following sub-sections.

**Dose-response functions.** To shorten the dose-calculation time, time-consuming radiation transport simulations in the environment were performed in advance, and the results were compiled into the dose-response functions. Here, the dose-response function means the dose contribution from a radiation source with unit radioactivity for a specific radionuclide to the target receptor at the ground level in the environment. The numerical values of the dose-response functions of $\dot{H}^*(10)$ and $\dot{K}_{Air}$ were evaluated for the radionuclides [85]Kr, [132]Te, [131]I, [132]I, [133]I, [133]Xe, [134]Cs, [136]Cs, and [137]Cs using PHITS and the dose-conversion coefficients [19] provided by the joint task group of the International Commission on Radiological Protection and the International Commission on Radiation Units and Measurements. The [137]Cs data include the contribution of gamma rays emitted from [137m]Ba in radioactive equilibrium with [137]Cs, and there are no [85]Kr and [133]Xe data for the ground source because those elements are noble gases with very low chemical reactivity. The units of the response functions of $\dot{H}^*(10)$ were given in units of mSv/h per $kBq/m^2$ for the ground sources and in units of mSv/h per $kBq/m^3$ for the cloud sources. The units of the response functions of $\dot{K}_{Air}$ were given in units of mGy/h per $kBq/m^2$ and mGy/h per $kBq/m^3$ for the ground and cloud sources, respectively.

**Fig 2** shows a simulation geometry for evaluating the dose-response functions by PHITS in the environment. Although PHITS simulated the interactions and transport for gamma rays and electrons, only gamma rays including bremsstrahlung were scored at the target receptor to convert gamma-ray fluences to doses using the conversion coefficients. The geometry consists of a right circular cylinder with a radius of 1000 m; the cylinder was considered to be infinite in extent [20, 21] and contained layers of air and soil with thicknesses of 1000 m and 1 m, respectively. The air–ground interface was assumed to be a flat surface. The target receptor with an area of $1 \times 1\ m^2$ was set in the center of the geometry 1 m above the ground. Ground sources distributed uniformly within the $1 \times 1\ m^2$ area with radioactivity of 1 kBq were placed on the ground surface inside a square domain from −500.5 m to 500.5 m centered on the target receptor. We assumed the condition of just after radionuclide descent from the atmosphere onto the ground and did not consider ground roughness and initial migration into the ground. The cloud sources, whose volume and activity were $1 \times 1 \times 1\ m^3$ and 1 $kBq/m^3$, respectively, were located in the air at altitudes of 1, 5, 10, 20, 30, 40, 50, 60, 80, 100, 125, 150, 200, 350, 500, 750, and 1000 m. We used the reciprocal transform method with the receptor and source [22,

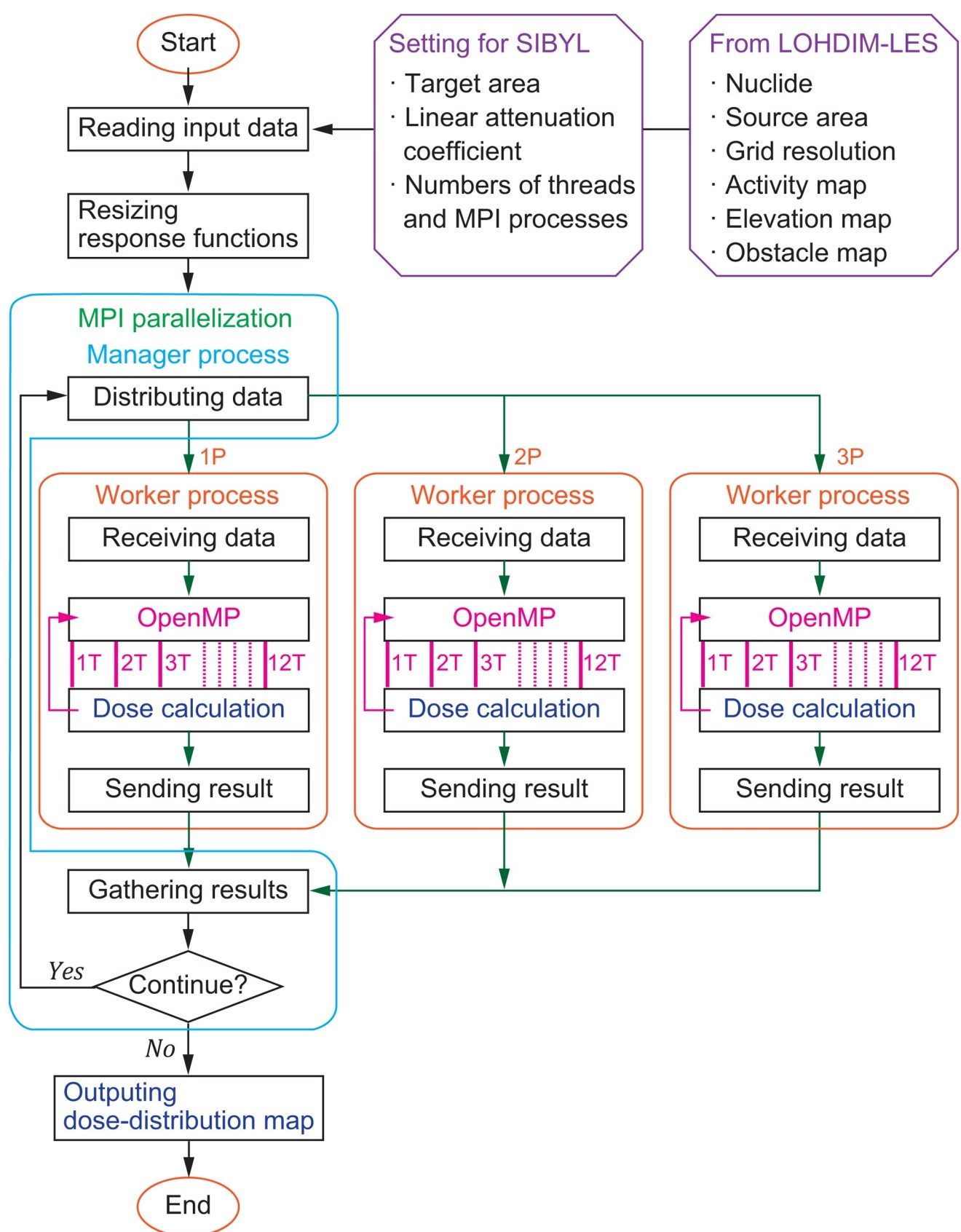

**Fig 1. Computational flow of SIBYL.** Hybrid parallel computation using 12 threads in open multi-processing (OpenMP) and 3 message-passing interface (MPI) worker processes are illustrated as an example. The numbers with the letters T and P indicate the identification numbers of the OpenMP threads and MPI worker processes, respectively.

23] in this process to obtain the results for all sources at once. A previous study [24] reported that the size of the dose-response function was sufficient to consider dose contributions from distant sources of the radionuclides. Additionally, we emphasize that the present dose-response functions include dose contributions from both primary and secondary gamma rays scattered in the air and soil.

Fig 3 shows an example of the response functions of $\dot{H}^*(10)$ for cloud sources of $^{137}$Cs. The responses decreased with increasing altitude and horizontal distance from the source to the target. Clearly, the responses at the altitude of 10 m become smaller than those at 100 m for distant sources that are approximately 250 m away from the center. This is because gamma rays emitted downward from distant sources at low altitudes tend to be blocked by the soil before reaching the target receptor.

**Dose-estimation procedure.** The procedure for calculating the dose-rate distribution is described in Fig 4. The area enclosed by the outermost line indicated by (A) in Fig 4 is the area of the radiation source. A Cartesian grid system that is identical to the grid system of LOH-DIM-LES is created inside the area (A) on the three-dimensional Cartesian coordinates of the $x$ and $y$ horizontal axes and the $z$ vertical axis. The data of the radioactivity concentrations of the radionuclide outputted by LOHDIM-LES are assigned to each cell of the grid. SIBYL defines the target area (indicated by (B) in Fig 4) to calculate the dose-rate distribution at the ground level using the same grid resolution as that of the radiation-source area. As an example, we focus on one cell, indicated by (C), at the $(x_i, y_j)$ position in the target area, indicated by

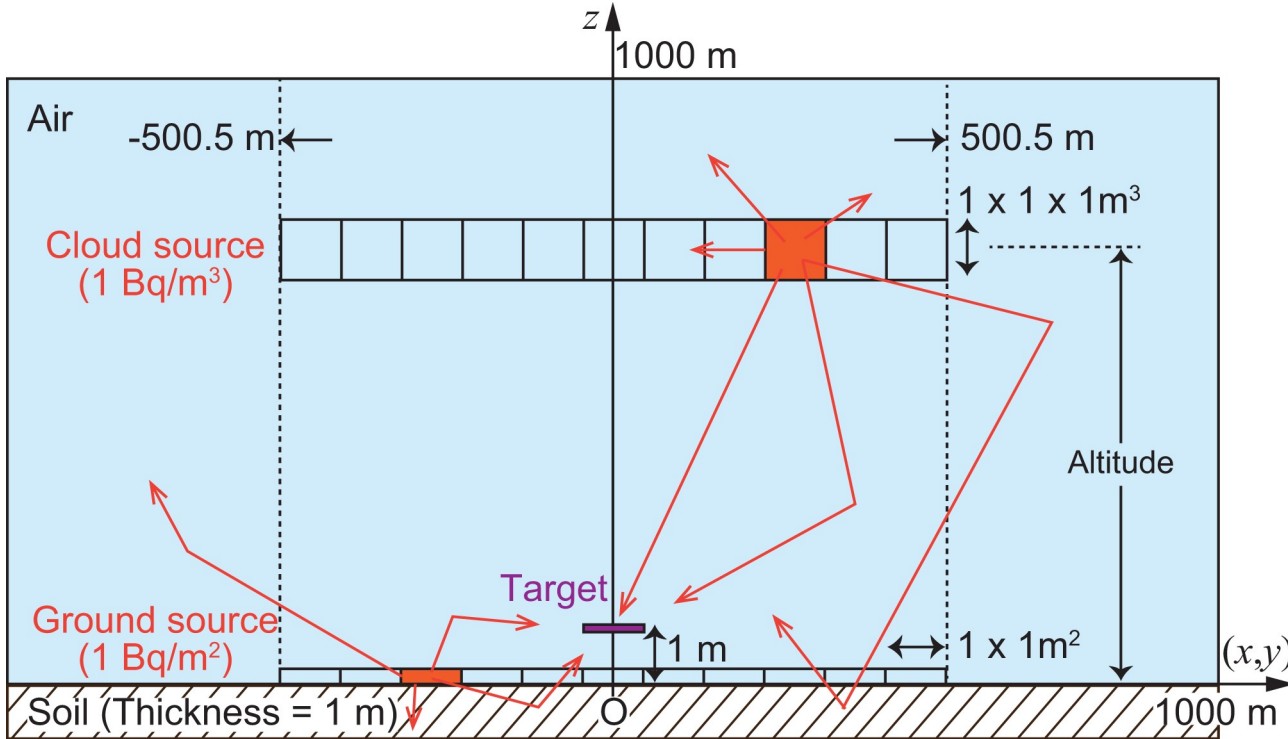

**Fig 2. Schematic of the simulation geometry of PHITS for evaluating dose-response functions.**

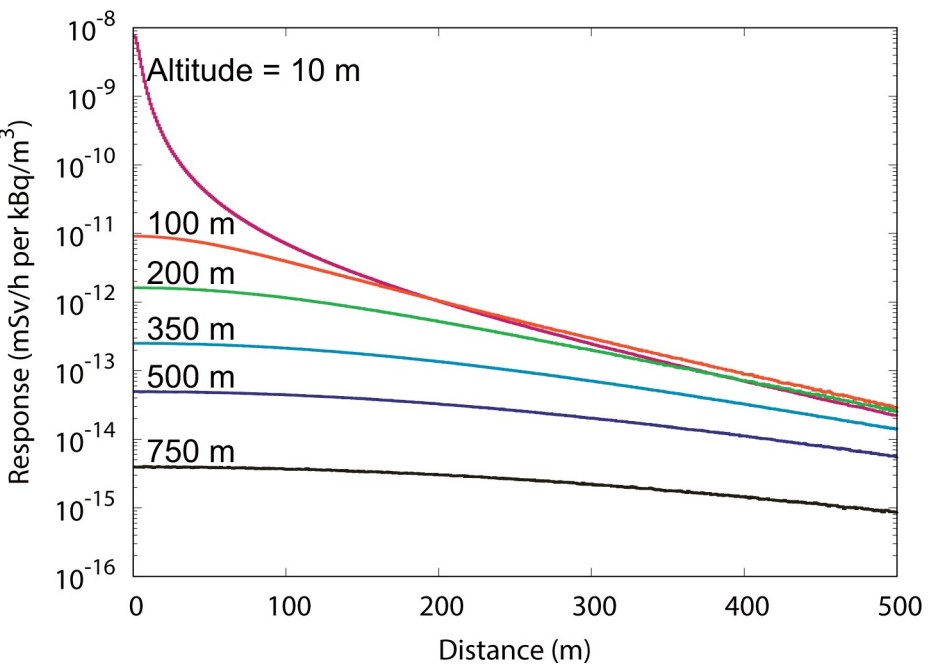

**Fig 3. Response functions of the ambient dose equivalent rate $\dot{H}^*(10)$ for cloud sources of $^{137}$Cs at the altitudes of 10, 100, 200, 350, 500, and 750 m above the ground.**

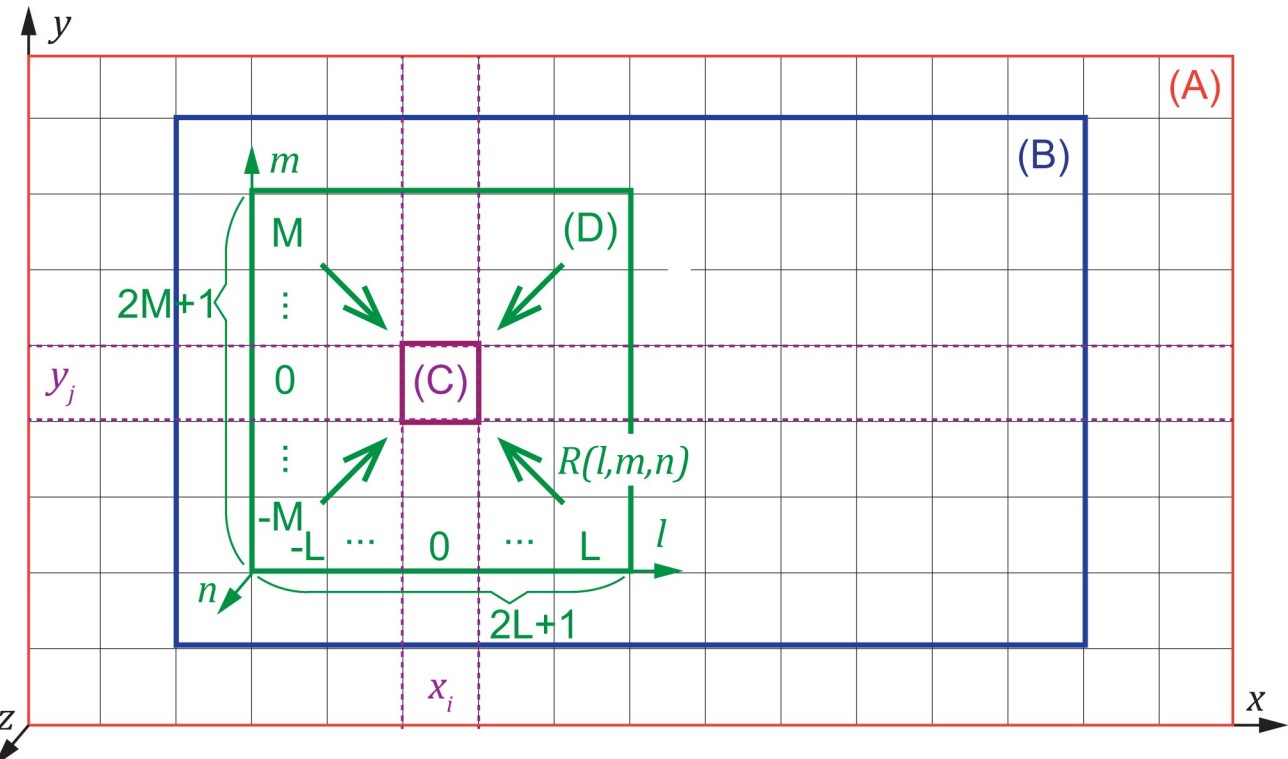

**Fig 4. Schematic of the areas treated in the dose calculation by SIBYL.** (A) Source area containing the radioactive plume and contaminated ground, (B) target area for calculating dose-rate distribution, (C) one cell of the grid established on the target area, and (D) the region of response functions considering the dose contribution from the source nuclides to the cell (C).

(B). To calculate the dose rate at (C), the dose-response functions are arranged centering on (C), as indicated by (D) in Fig 4. The spatial resolution of the original dose-response functions is 1 m inside a 1001 m × 1001 m horizontal domain (as described above), and they are resized to coincide with the resolution of the source and target areas. The numerical values of the resized dose-response functions are computed using an arithmetic mean of data from corresponding elements of the original dose-response functions. Additionally, the values along the $z$ axis are obtained by logarithmic interpolation based on the original data at specific altitudes. The region of the dose-response functions is divided into $2L + 1$, $2M + 1$, and $N$ number of cells with respective resized resolutions in the $l$ and $m$ horizontal axes and $n$ vertical axis on the Cartesian coordinate system of those response functions (Fig 4). The region of the dose-response functions is truncated if there are division remainders. The external gamma-ray dose rate delivered to (C) at the $(x_i, y_j)$ position is calculated as follows:

$$\dot{D}(x_i, y_j) = \sum_{l=-L}^{L} \sum_{m=-M}^{M} \sum_{n=0}^{N} R(l, m, n) \cdot C(x_i + l, y_j + m, n) \quad (1)$$

where $\dot{D}\left(x_i, y_j\right)$ [mSv/h] is the dose rate at $(x_i, y_j)$; $R(l, m, n)$ [mSv/h per kBq/m$^2$ for $n = 0$ and mSv/h per kBq/m$^3$ for $n \neq 0$] is the dose-response function in the $(l, m, n)$ coordinate system; and $C(x_i + l, y_j + m, n)$ [kBq/m$^2$ for $n = 0$ and mSv/h per kBq/m$^3$ for $n \neq 0$] is the activity concentration in the cell for the $(x, y, z)$ coordinates corresponding to the $(l, m, n)$ cell.

By using pre-calculated dose-response functions, SIBYL can skip the time-consuming radiation-transport simulation in the environment and calculate dose rates at the ground-level cells quickly and accurately to apply to emergency responses. To increase the reading speed, the data of dose-response functions were stored as 8-byte floating-point values in unformatted binary stream files.

**Attenuation of doses by obstacles.** To consider gamma-ray dose attenuation due to obstacles inside the calculation domain, Eq (1) is modified as follows:

$$\dot{D}'(x_i, y_j) = \sum_{l=-L}^{L} \sum_{m=-M}^{M} \sum_{n=0}^{N} R(l, m, n) \cdot C(x_i + l, y_j + m, n) \cdot \exp(-\mu L_{s \to t}) \quad (2)$$

where $\dot{D}'\left(x_i, y_y\right)$ [mSv/h] is the dose rate at $(x_i, y_j)$ considering dose attenuation due to obstacles; $\mu$ [m$^{-1}$] is the linear dose-attenuation coefficient of the obstacles; and $L_{S \to T}$ [m] is the total length of the obstacles in a straight line from the center of the source cell at $(x_i + l, y_j + m, n)$ to the center of the target cell at $(x_i, y_j)$.

The current version of SIBYL limits obstacle composition to one material type in one simulation, and its linear dose-attenuation coefficient is set by users as a constant parameter that reflects the following: (i) the average energy of gamma rays emitted from the radionuclides, (ii) the total attenuation coefficient for the gamma rays at that energy, and (iii) the density of the obstacles. This means that the energy degradation and buildup of gamma rays passing through obstacles were not considered in the dose-attenuation process, whereas the interactions of the gamma rays in the atmosphere were included in the dose-response functions. To improve speed, SIBYL skips dose calculations at target cells with obstacles.

**Elevation at the target cells.** Fig 5 illustrates how terrain elevation data are used for dose calculations. This function was designed to estimate cloud-shine doses from an overhead radioactive plume, and thus is not applicable to ground-shine dose calculations from contaminated ground. In addition, radiation shielding by elevated soil was not considered even in the cloud-shine calculations. As mentioned in the dose-estimation procedure section, SIBYL

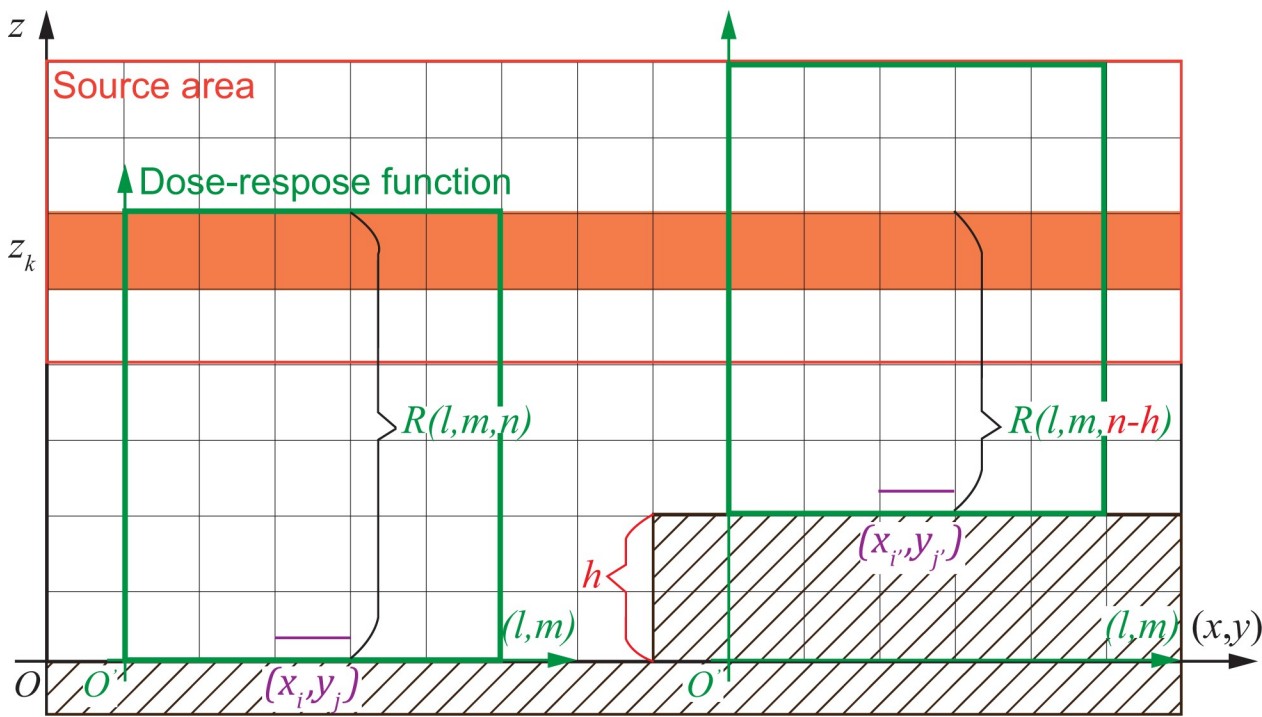

**Fig 5. Schematic of the use of terrain elevation data in SIBYL.** The elevation at the cell of $(x_{i'}, y_{j'})$ is $h$ cells higher than at $(x_i, y_j)$.

calculates dose rates at 1 m above the terrain surface. The data sets of the terrain elevation were prepared in meters for the target area and converted to multiple vertical cells on the grid system depending on grid resolutions. The standard level of elevation was set at the lowest elevation in the target area, and elevation differences from the standard level were expressed as a positive number of cells. The dose rate at the target cells with elevation data was estimated using Eq (1) by considering the offset of elevation in the selection of dose-response functions:

$$\dot{D}_E(x_{i'}, y_{j'}) = \sum_{l=-L}^{L} \sum_{m=-M}^{M} \sum_{n=h+1}^{N+h} R(l, m, n-h) \cdot C(x_{i'} + l, y_{j'} + m, n) \qquad (3)$$

where $\dot{D}_E\left(x_{i'}, y_{j'}\right)$ [mSv/h] is the dose rate at the target cell of $(x_{i'}, y_{j'})$ with an elevation of $h$ cells.

**Design of parallel computing algorithm.** The parallel algorithm of SIBYL was designed for shared-memory and distributed-memory communications based on open multi-processing (OpenMP) and message-passing interface (MPI) technologies, respectively. Hybrid parallel computing is also executable on a computer cluster using both OpenMP and MPI, as shown in **Fig 1**. After reading the input data and resizing the dose-response functions, the algorithm divides the target area into small groups. The MPI manager process distributes tasks corresponding to those small groups to each worker process for MPI-based parallel computation. When the worker process finishes the calculations for that task, the manager process gathers the result and then distributes a new task to that worker process. This procedure balances the computational burden among the worker processes. Furthermore, a summing loop for the dose calculation expressed in Eqs (1)–(3) can be processed using multithreading on OpenMP technology. The code divides the dose-calculation task among the threads, which then run

concurrently. The results for the target groups gathered in the manager process are combined to create a map of the dose distribution inside the target area.

In this study, we estimated the performances of parallel computation by SIBYL on the JAEA computer cluster system (SGI ICE X, manufactured by Hewlett-Packard Enterprise). This system was based on Intel Xeon E5-2680 v3 processors (each with 12 cores with a 2.5 GHz base frequency) and a 4X fourteen data rate InfiniBand network with dual-plane hyper-cube technology. Each computing node of the system had two processors and 64 GB of main memory. The compiler was the Intel Fortran compiler 18.0.3 with the SGI Message-Passing Toolkit, and the code was compiled on SUSE Linux Enterprise Server 11 SP3 using the compiler options *-fpp -O3 -xCORE-AVX2 -no-prec-div -fp-model fast = 2 -align array64byte*.

To express the parallel computation performance, the speed-up factor $S$ and parallel efficiency $\varepsilon$ were defined as follows:

$$S = \frac{T_{\text{Serial}}}{T_{\text{Parallel}}},$$

$$\varepsilon = \frac{s}{N_{\text{Process}}} \tag{4}$$

where $T_{\text{Serial}}$ and $T_{\text{Parallel}}$ are the elapsed time of the serial and parallel computations, respectively, and $N_{\text{Process}}$ is the number of processing elements used in parallel computation.

## Case setup

To examine the validity of SIBYL calculations, we investigated five typical cases for steady-state plume dispersion over terrain and unsteady-state plume dispersion in the building arrays. We then compared these results with those obtained using PHITS. The details of the conditions set in each case are described below.

**Case study for a steady-state Gaussian plume dispersion.** Table 1 summarizes the cases (1A, 1B, and 1C) examined in this study for steady-state plume dispersion. We supposed a hypothetical release of [85]Kr radioactive gas from a ventilation shaft under steady atmospheric conditions. The Gaussian plume model (GPM) [25, 26] was employed in those cases as a dispersion model instead of LOHDIM-LES because GPM's simple analytical solution had been used in prior research as a standard approach for studying the air pollutant dispersion under steady conditions.

Table 1. Case studies for a steady-state Gaussian plume.

| Case | 1A | 1B | 1C |
|---|---|---|---|
| Radiation source | Steady-state plume | | |
| Radionuclide | [85]Kr | | |
| Dispersion model | Gaussian plume model (GPM) | | |
| Resolution | 5 m | 100 m | |
| Target area | $-1 \text{ km} \le x \le 1 \text{ km}$ | $-1 \text{ km} \le x \le 10 \text{ km}$ | |
| | $-1 \text{ km} \le y \le 1 \text{ km}$ | $-10 \text{ km} \le y \le 10 \text{ km}$ | |
| Geometry | Flat terrain | | Hilly terrain |
| Dose model | SIBYL | | |
| | PHITS | | |

Fig 6 illustrates the dispersion of radioactive gas in the GPM; which can be expressed as follows:

$$C(x, y, z) = \frac{Q}{2\pi\sigma_y(x)\sigma_z(x)U} \cdot \exp\left(-\frac{y^2}{2\sigma_y(x)^2}\right) \cdot \left[\exp\left(-\frac{(z-H)^2}{2\sigma_z(x)^2}\right) + \exp\left(-\frac{(z+H)^2}{2\sigma_z(x)^2}\right)\right] \quad (5)$$

where $C(x, y, z)$ [Bq/m$^3$] is the activity concentration on the $x$, $y$, and $z$ [m] Cartesian coordinates that specify the downwind distance from a release point, the crosswind distance from the emission plume centerline, and the vertical direction above the ground, respectively. $Q$ [Bq/s] represents the emission rate of the radioactive material, and $U$ [m/s] shows the mean wind speed along the plume centerline. $\sigma_y(x)$ and $\sigma_z(x)$ [m] are the parameters that depend on the $x$ coordinate and represent the standard deviations of the centerline of the Gaussian distributions in the $y$ and $z$ directions, respectively. $H$ [m] is the height of the emission plume centerline above the ground. The $\sigma_y(x)$ and $\sigma_z(x)$ values were taken from the Pasquill–Gifford curve [27]. As shown in Eq (5), GPM assumes Gaussian distributions for the crosswind and vertical dispersions of the plume and the effect of ground reflection. In this study, the parameters $Q$, $U$, and $H$ were set to 1 Bq/s, 1 m/s, and 150 m, respectively, for the emission of $^{85}$Kr gas.

Case 1A established the target area of the gamma-ray dose-rate calculation in a 2-km square centered on the origin of the $(x, y, z)$ coordinate system with a 5-m grid resolution and at 1 m above the ground. The target areas of cases 1B and 1C were defined to be wider than the target area of case 1A. The target area ranged from −1 km to 10 km in the $x$ direction and from −10 km to 10 km in the $y$ direction, with a grid resolution of 100 m. In addition, the ground was flat for cases 1A and 1B, whereas the terrain of case 1C was elevated to 10 m and 30 m in the ranges of 2 km $\leq x <$ 4 km and 4 km $\leq x \leq$ 10 km, respectively, as compared with the ground

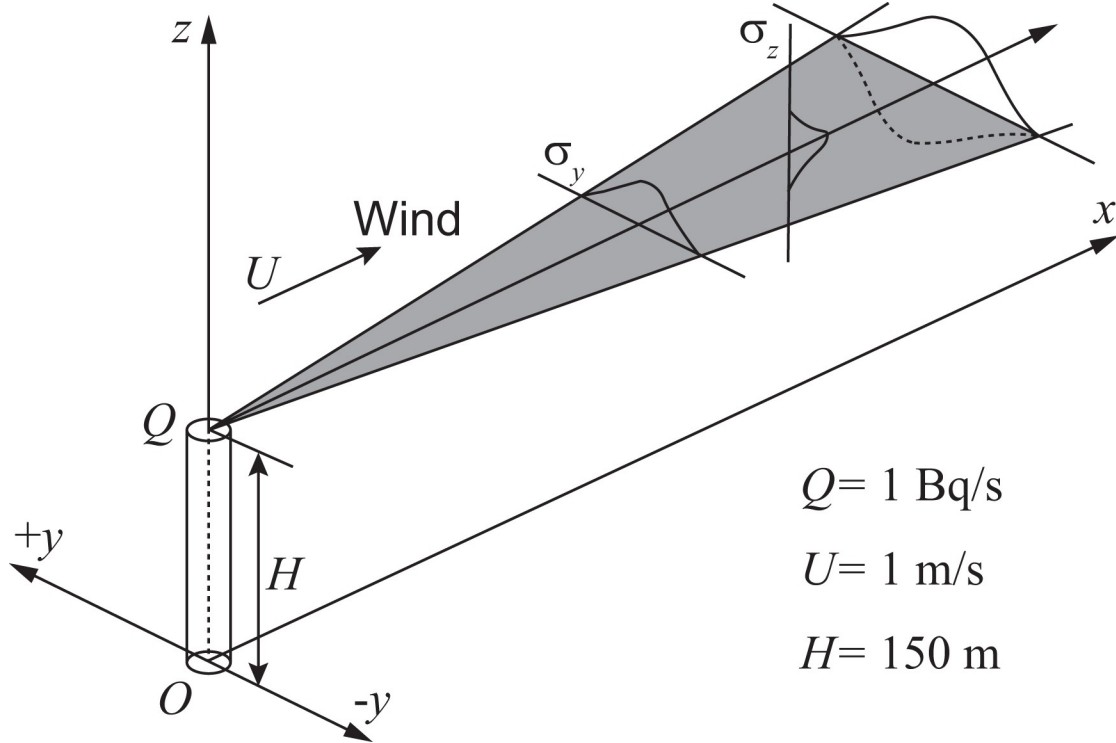

**Fig 6. Schematic of a steady-state Gaussian plume for cases 1A, 1B, and 1C.**

level of $x < 2$ km (see **Fig 7**). The ground elevations were homogeneous with respect to the $y$ direction in this case.

The dose-rate distributions were calculated by SIBYL and PHITS on the grid system. The activity concentrations given by GPM were integrated within a cell of the grid and then inputted to SIBYL and PHITS as the grid data of radiation source. The output data of the dose rates were normalized to the area of the target cells.

**Case study for an unsteady-state dispersion in building arrays.**   **Table 2** lists cases 2A and 2B for an unsteady dispersion of the $^{137}$Cs pollutant and its deposition, respectively, in building arrays. The activity concentrations of the pollutant in the air and on the surface of the geometry were provided by LOHDIM-LES. The upper and lower panels of **Fig 8** depict the distributions of the activity concentrations in cases 2A and 2B, respectively.

The size of the simulation geometry was set to 240 m × 240 m × 150 m on the three-dimensional Cartesian coordinates of the $x$ and $y$ horizontal axes and the $z$ vertical axis, whose origin was the center of the horizontal plane. The air–ground interface was set at $z = 0$. The region was segmented by grids with a 1-m spatial resolution in the horizontal direction and a 1–4-m spatial resolution along the vertical direction. The cubic buildings were arranged in five rows and five columns in a regular square array over flat terrain. Each building measured 24 m × 24 m × 24 m and was assumed to be composed of a mixture of air (90%) and concrete (10%). The density of the buildings was set to 0.24 g/cm$^3$, and the linear dose-attenuation coefficient used as an input parameter in the SIBYL calculation was determined to be 1.85 m$^{-1}$ using the PHITS code for the $^{137}$Cs source.

The point of the hypothetical release of the $^{137}$Cs dispersion simulation was set at (−36, 0, 0), which was just in front of a building and on the ground. The simulation along time was performed under the following conditions: (i) the total amount of radioactivity was 347.25 Bq, (ii) the air flow was driven by the pressure gradient along the $x$ axis from negative to positive, (iii)

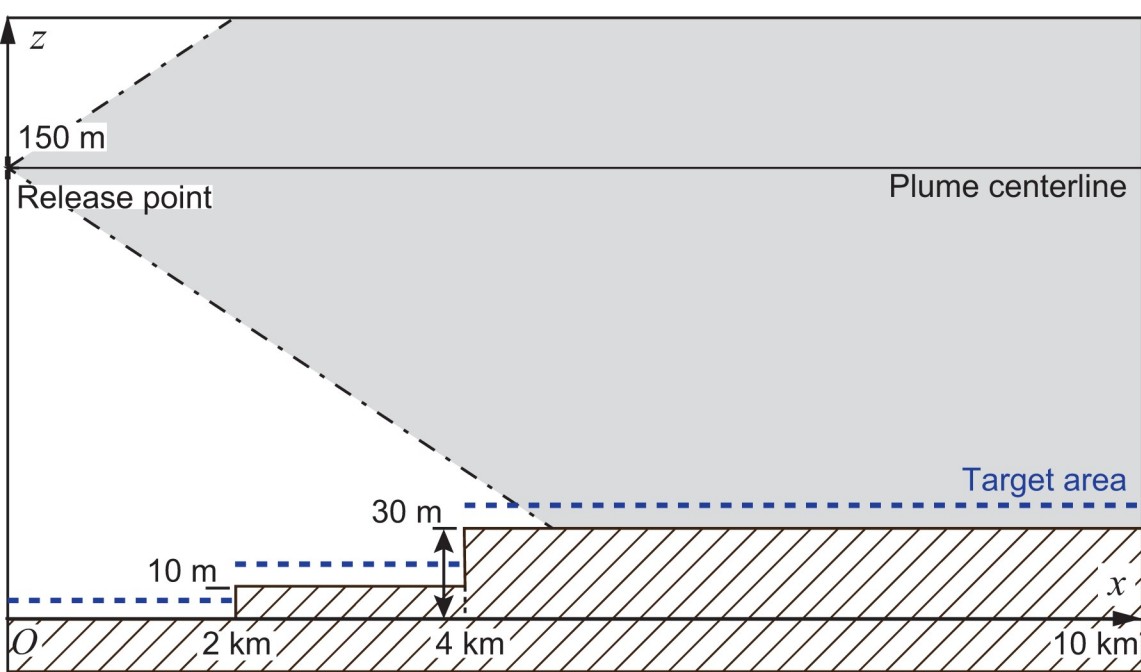

**Fig 7. Schematic of a cross-sectional view for the elevated terrain modeled for case 1C.** The dashed line indicates the target cells set at 1 m above the ground.

**Table 2. Case studies for an unsteady plume dispersion in building arrays.**

| Case | 2A | 2B |
|---|---|---|
| Radiation source | Unsteady-state plume | Surface contamination |
| Radionuclide | $^{137}$Cs | |
| Dispersion model | LOHDIM-LES | |
| Resolution | 1 m | |
| Target area | $-120 \text{ m} \leq x \leq 120 \text{ m}$ | |
| | $-120 \text{ m} \leq y \leq 120 \text{ m}$ | |
| Geometry | Building arrays on flat terrain | |
| Dose model | SIBYL | |
| | PHITS | |

the deposition rates were given as 0.05 cm/s and 0.1 cm/s for the building and ground surfaces, respectively, and (iv) the time step for LES was set to 0.01 s. Case 2A reproduced the instantaneous distribution of $^{137}$Cs in the air 12 min after the initial release, and case 2B reproduced the instantaneous distribution for the deposition of $^{137}$Cs on the ground and building surfaces 30 min after release.

The grid data of the activity concentration given by LOHDIM-LES were inputted to SIBYL and PHITS, and the distribution map of the external gamma-ray doses were calculated inside the target area considering dose attenuation due to buildings. These results were normalized to the area of the target cells.

## Results and discussion

This section gives the results of the external gamma-ray dose distributions calculated by SIBYL and PHITS for cases 1A, 1B, and 1C (which correspond to the steady-state Gaussian plume dispersion predicted by GPM) and for cases 2A and 2B (which correspond to the unsteady-state dispersion in the building arrays by LOHDIM-LES). The gamma-ray doses show ambient dose equivalent rates at 1 m above the ground for each target cell on the grid. The statistical uncertainties of the Monte Carlo simulation by PHITS were given as one standard deviation, and in most cells in the target area these uncertainties were below 1%.

### Steady-state Gaussian plume dispersion

**Fig 9** shows the distribution map of the ambient dose equivalent rate calculated by SIBYL and PHITS for case 1A, that is, the local-scale Gaussian plume dispersion of $^{85}$Kr within 1000 m in the downwind direction with a 5-m spatial resolution. An area of dose rates greater than $1 \times 10^{-12}$ μSv/h was observed from the point $(x, y) = (0, 0)$, which was directly underneath the hypothetical release point, toward the leeward side along the plume centerline of $y = 0$ m. The dose rates gradually decreased by separation from the centerline in the crosswind direction. **Fig 10** shows the dose rates calculated by SIBYL and PHITS along the lines $y = 0$ m and $x = 500$ m. These results showed good agreement, with differences of up to 3%. The dose rates in the upwind region of $x < 0$ were the result of $^{137}$Cs radioactivity distributed in the leeward side at $x \geq 0$. Compared to the rates at the release point, these rates were decreased by approximately one order of magnitude at $x = -200$ m.

**Fig 11** shows the dose-distribution map for case 1B. In this case, the target area was enlarged to 10 km in the downwind direction with a coarse spatial resolution of 100 m. As in case 1A, target cells with high dose rates were observed along the plume centerline. The dose distributions on lines $y = 0$ km and $x = 5$ km calculated by SIBYL were compared with those of

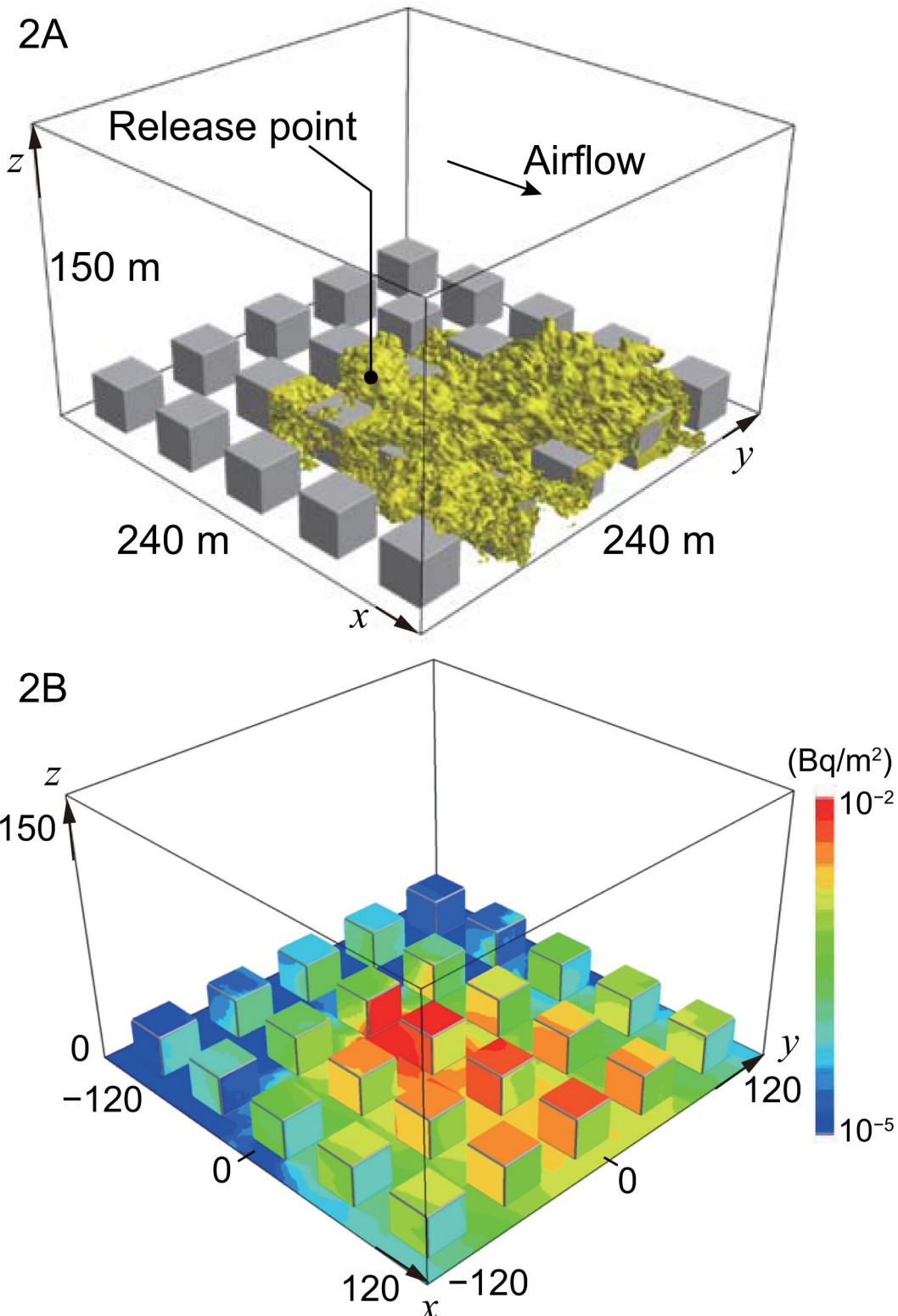

**Fig 8. Distribution maps of activity concentrations in building arrays simulated by LOHDIM-LES.** The upper panel represents the atmospheric dispersion of [137]Cs for case 2A. The yellow areas on the iso-surface indicate 0.01% of the initial concentration. The lower panel depicts the deposition of [137]Cs on the ground, building walls, and roofs for case 2B.

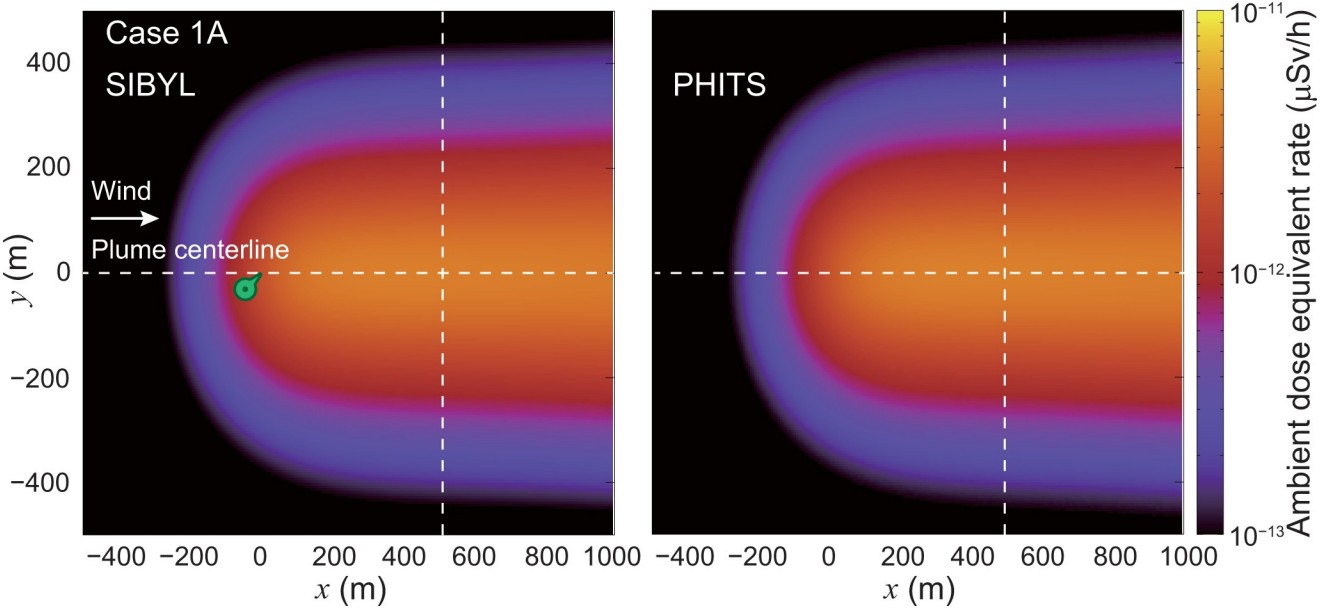

**Fig 9. Distribution map of ambient dose equivalent rate at 1 m above the ground on the grids of the target area for case 1A.** The left and right panels show the calculation results of SIBYL and PHITS, respectively. The dashed lines indicate the lines $y = 0$ m and $x = 500$ m. The marker on the map represents the location underneath the hypothetical release point.

PHITS in **Fig 12**. The results of SIBYL differed from those of PHITS by up to 5% at the target cells under the plume centerline with a crosswind distance of less than 0.5 km from the centerline. For the cells more than 0.5 km away from the plume centerline, SIBYL slightly underestimated the dose rates. The deviation between SIBYL and PHITS increased up to 20% with increasing crosswind distance from the centerline. The doses even at the distant cells from the plume centerline were affected by radionuclides distributed around the centerline. SIBYL failed to reproduce the dose contribution to cells more than 0.5 km from the source because the size of the dose-response function of SIBYL was approximately 0.5 km in one direction. However, the absolute values of those dose contributions were quite small; therefore, we concluded that the size of the dose-response function was appropriate with good calculation speed and reasonable accuracy.

**Fig 13** shows the calculation results of SIBYL and PHITS on the line $y = 0$ km in the target area for case 1C. This case is the same as case 1B but with elevated terrain, as shown in **Fig 7**. SIBYL reproduced the increase of the dose rates in the elevated cells due to decreased vertical distance between the target cells and the overhead plume. The results of SIBYL differed from those of PHITS by up to 5% in most of the target cells. The dose rates of PHITS dropped in the cells immediately in front of the ground uplift at $x = 2$ km and $x = 4$ km because cloud shine from the overhead plume was blocked by the elevated terrain. Although SIBYL could not consider the effect of radiation shielding by the soil, the deviation was not significant and was less than 7%.

From these results, we concluded that SIBYL estimated ground-level dose rates with sufficient accuracy for steady-state plume dispersion over flat and elevated terrain inside a calculation domain of up to 10 km.

## Unsteady-state dispersion in building arrays

**Fig 14** mapped the distribution of the ambient dose equivalent rate calculated by SIBYL and PHITS for the unsteady-state dispersion of $^{137}$Cs in the building arrays established in case 2A.

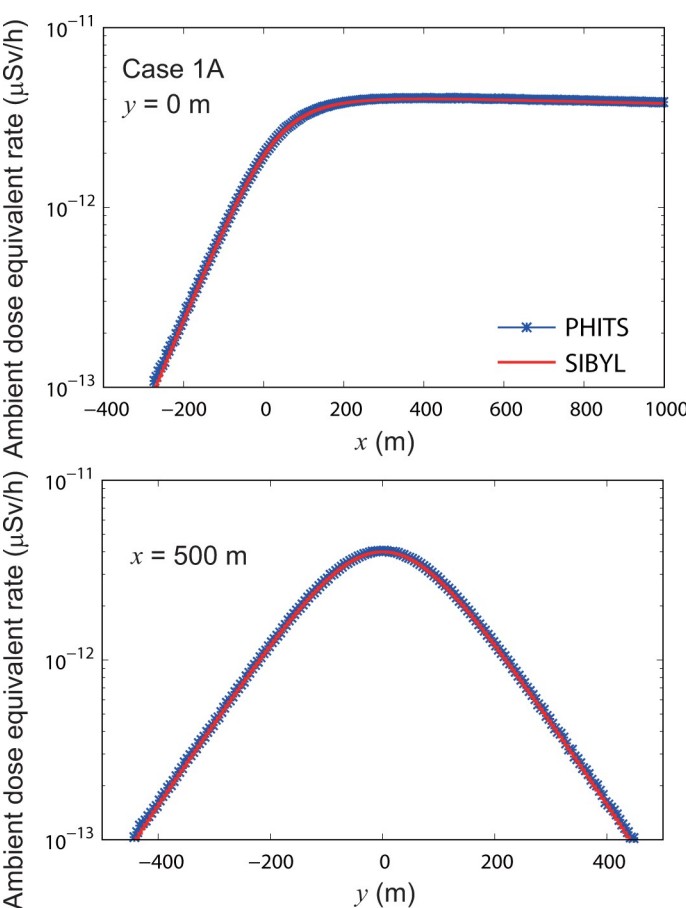

**Fig 10. Distributions of ambient dose equivalent rates on specific lines in the target area for case 1A.** The upper panel indicates the calculation results of SIBYL and PHITS on the line $y = 0$ m. The lower panel represents the calculation results on the line $x = 500$ m.

As mentioned, SIBYL did not calculate the doses in cells occupied by the obstacles. For the most part, both SIBYL and PHITS provided distribution maps on the same level of dose rates. However, the silhouettes of the low-dose areas behind buildings calculated using SIBYL are sharper than those calculated using PHITS. For a quantitative comparison, we compared the results of SIBYL with those of PHITS on the lines $y = 0$ m and $x = −25$ m in **Fig 15**. The results of SIBYL differed from those of PHITS by up to 10% in the leeward region from the release point (−36, 0, 0). These discrepancies are acceptably small; however, SIBYL yielded systematically smaller values than PHITS. Furthermore, on the windward side in $−84$ m $< x < −60$ m behind the buildings, SIBYL underestimated doses by a factor of four as compared with the PHITS results.

To analyze the reasons for the underestimations on the leeward and windward sides, we performed PHITS calculations by substituting obstacles made of concrete with air and an ideal radiation absorber. **Fig 16** shows the results of PHITS for case 2A on the line $y = 0$ m with obstacles of air and the ideal absorber together with the results of SIBYL shown on the upper panel of **Fig 15**. The ideal radiation absorber is a material that completely absorbs all radiation. Compared to the PHITS results calculated by replacing the obstacles with air, PHITS provided much larger values than SIBYL. This result assures that the dose-attenuation algorithm introduced into SIBYL worked effectively. On the leeward side of $x > −36$ m, the results of SIBYL

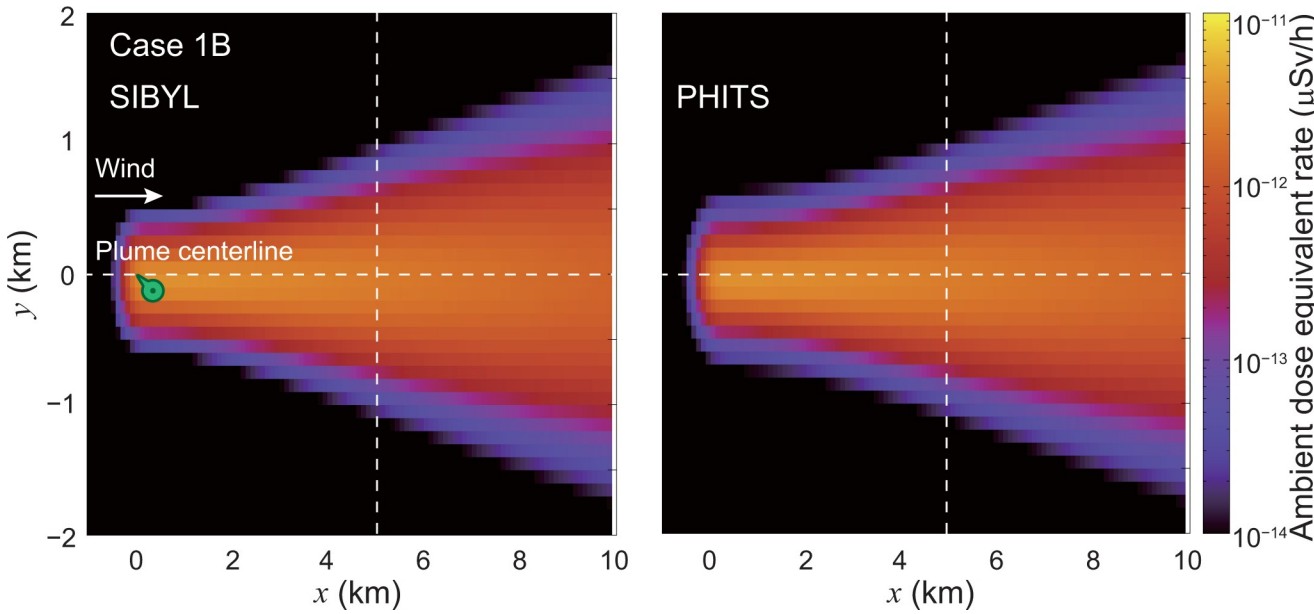

**Fig 11. Distribution map of ambient dose equivalent rate at 1 m above the ground on the grids of the target area for case 1B.** The dashed lines indicate the lines $y = 0$ km and $x = 5$ km. The marker on the map represents the location underneath the hypothetical release point.

differed from those of PHITS calculated with the ideal radiation absorber by up to 1%; however, there was a 10% deviation from the results of PHITS with concrete obstacles. This means that the 10% underestimation of SIBYL in the same region observed in **Fig 15** was because of gamma-ray scattering on building walls. SIBYL does not consider gamma-ray scattering on obstacles.

On the windward side $-84$ m $< x < -60$ m behind the buildings, the results of SIBYL were less than those of PHITS with the ideal absorber. The discrepancy originated from SIBYL's treatment of streaming gamma rays in the intricate geometry of the building arrays. While searching for obstacles on the path from the source to the target cells, SIBYL identified those cells by the center points without any spatial resolution of the cells. When obstacles existed on the straight line connecting the source and target points, SIBYL applied the attenuation algorithm even for streaming gamma rays that reached the target cell without passing through the obstacles by scattering in the atmosphere. This led to over-shielding of the gamma rays. However, the dose contributions by those streaming gamma rays were relatively low and are not significant in the dose estimations required during an initial response to a nuclear emergency.

Along the line $x = -25$ m on the lower panel of **Fig 15**, the dose rates calculated by SIBYL were approximately 10% greater than those calculated by PHITS in the region where the crosswind distance $y$ was greater than 100 m from the release point at $y = 0$ m. The dose-response functions used in SIBYL included dose contributions by secondary gamma rays generated in the environment, which was assumed to be infinite. For case 2A, a portion of the secondary gamma rays scattered by environmental media outside the straight line from a source cell around the release point to a distant target cell had to be shielded by the building array. Nevertheless, SIBYL could not consider the shielding of those gamma rays when the obstacles did not exist on the straight line connecting the source and target cells. Therefore, SIBYL slightly overestimated the dose rate at distant locations in this case.

**Fig 17** shows dose-distribution maps for case 2B calculated by SIBYL and PHITS. **Fig 18** shows the results of the comparisons on the lines $y = 0$ m and $x = -25$ m. On the leeward side

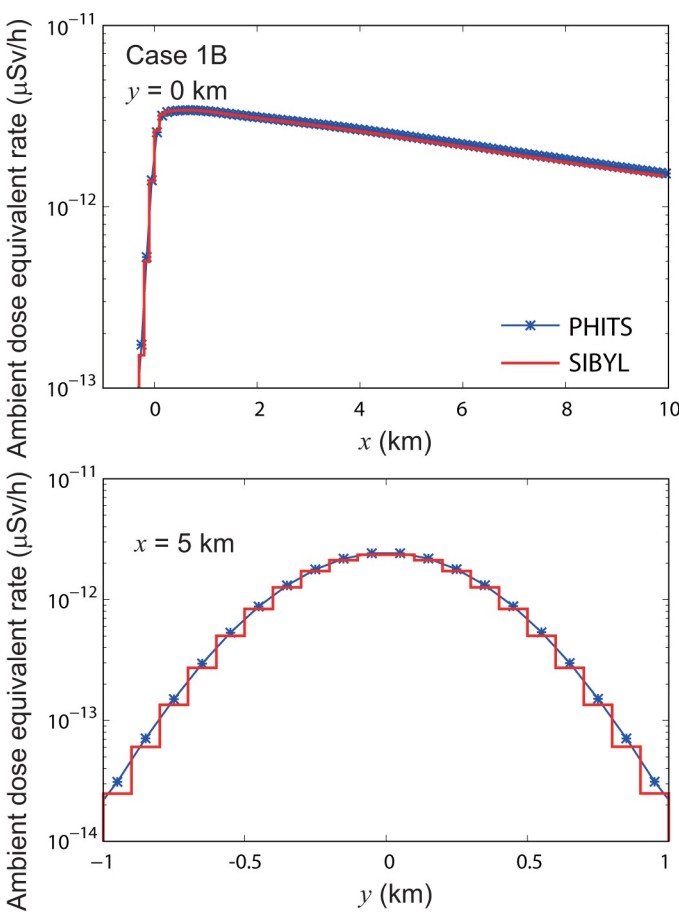

**Fig 12. Distributions of ambient dose equivalent rates on specific lines in the target area for case 1B.** The upper panel indicates the calculation results of SIBYL and PHITS on the line $y = 0$ km. The lower panel represents the calculation results on the line $x = 5$ km.

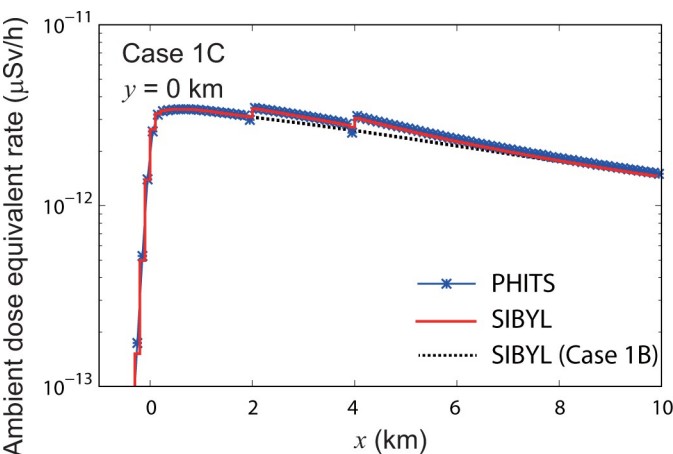

**Fig 13. Distribution of ambient dose equivalent rate for case 1C on the line $y = 0$ km.** The result of SIBYL for case 1B on flat terrain is also drawn with a dashed line.

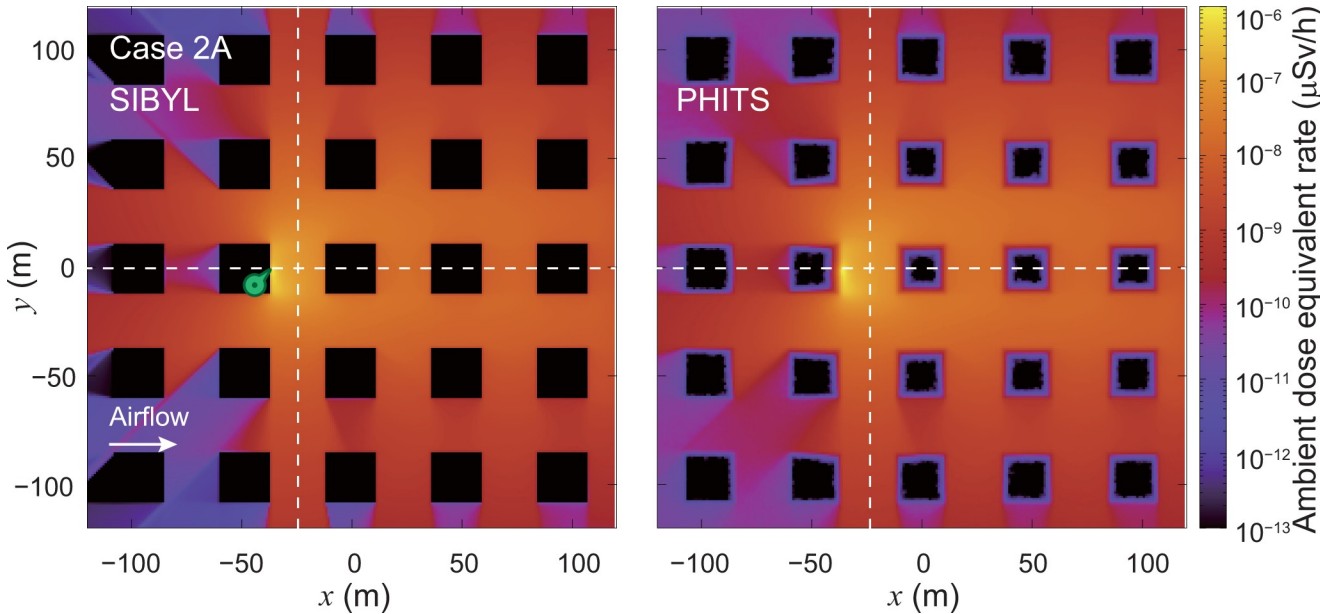

**Fig 14. Distribution map of ambient dose equivalent rate at 1 m above the ground on the grids of the target area for case 2A.** The left and right panels are the calculation results of SIBYL and PHITS, respectively. The dashed lines indicate the lines $y = 0$ m and $x = -25$ m. The marker on the map represents the hypothetical release point at $(-36, 0, 0)$.

in $x > -35$ m, the results of SIBYL differed by up to 3% from those of PHITS. Unlike dose contributions for case 2A, the dose contributions by gamma rays scattered on building walls were not significant because most of the radiation sources were on the ground, and the primary gamma-ray dose contributions to the target cells were dominant. On the windward side $-84$ m $< x < -60$ m behind the buildings, SIBYL underestimated the doses. These results were similar to those observed in case 2A. At more than 100 m from the hypothetical release point in the $y$ direction, SIBYL overestimated doses by a maximum of 20% because SIBYL estimated the dose contributions of secondary gamma rays in a semi-infinite atmosphere without obstacles using the dose-response functions. Our results demonstrate that SIBYL has a good ability to assess dose-distribution maps with a sufficient precision comparable to that of PHITS, especially for high-dose areas where primary gamma rays emitted from the radionuclide play an important role, even in cities with building arrays.

## Performance of parallel computation

**Table 3** summarizes the results of the performance evaluation of SIBYL's parallel computation for the case studies in this research. The values of $N_T \times N_R \times N_{Sz}$ are provided as an index to represent the size of the problem to be calculated, where $N_T$ is the number of cells in the target area and $N_R$ is the number of cells in the resized dose-response function in a horizontal plane. $N_{Sz}$ denotes the number of vertical cells inside the source region. The parallel computations were performed using 96 processing elements through a hybrid of 12 threads on OpenMP and 8 MPI worker processes.

Cases 1A, 1B, and 1C involved gamma-ray dose estimations for a steady-state Gaussian plume without obstructions. For case 1A, using 96 parallel processing elements yielded a speed-up factor of 70.77 and parallel efficiency of 0.74. However, in cases 1B and 1C, the parallel efficiencies were evaluated to be 0.03. This was because the parallelized dose-estimation routine did not take much time for the whole computation compared to other routines such as

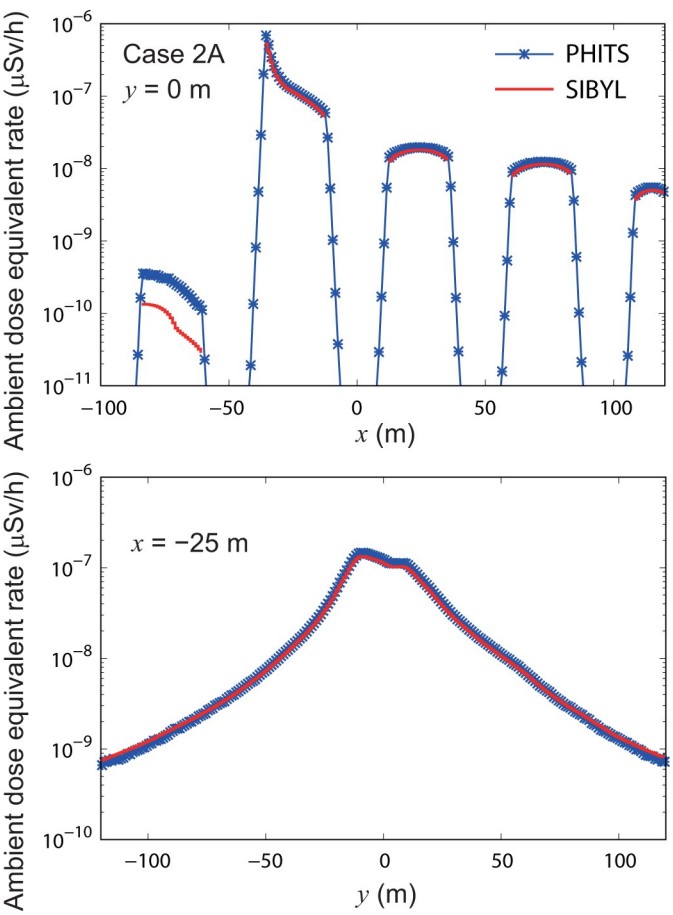

**Fig 15. Distributions of ambient dose equivalent rates on specific lines in the target area for case 2A.** The upper panel shows the calculation results of SIBYL and PHITS on the line $y = 0$ m. The lower panel represents the calculation results on the line $x = -25$ m.

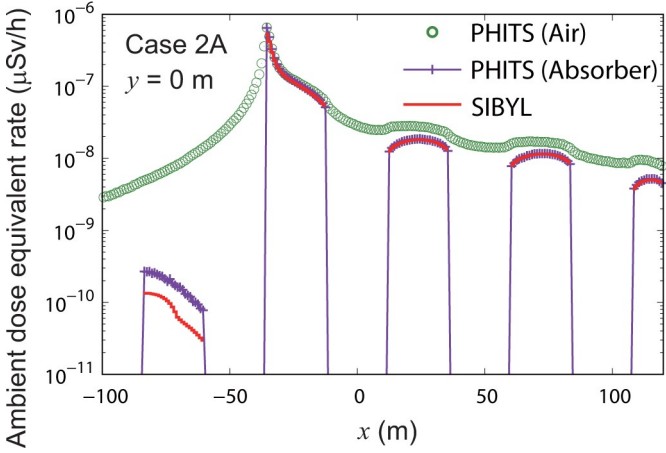

**Fig 16. Distribution of ambient dose equivalent rate for case 2A on the line $y = 0$ m.** The open circles and crosses indicate the calculation results of PHITS, which replaced the concrete obstacles of case 2A with air and an ideal radiation absorber, respectively.

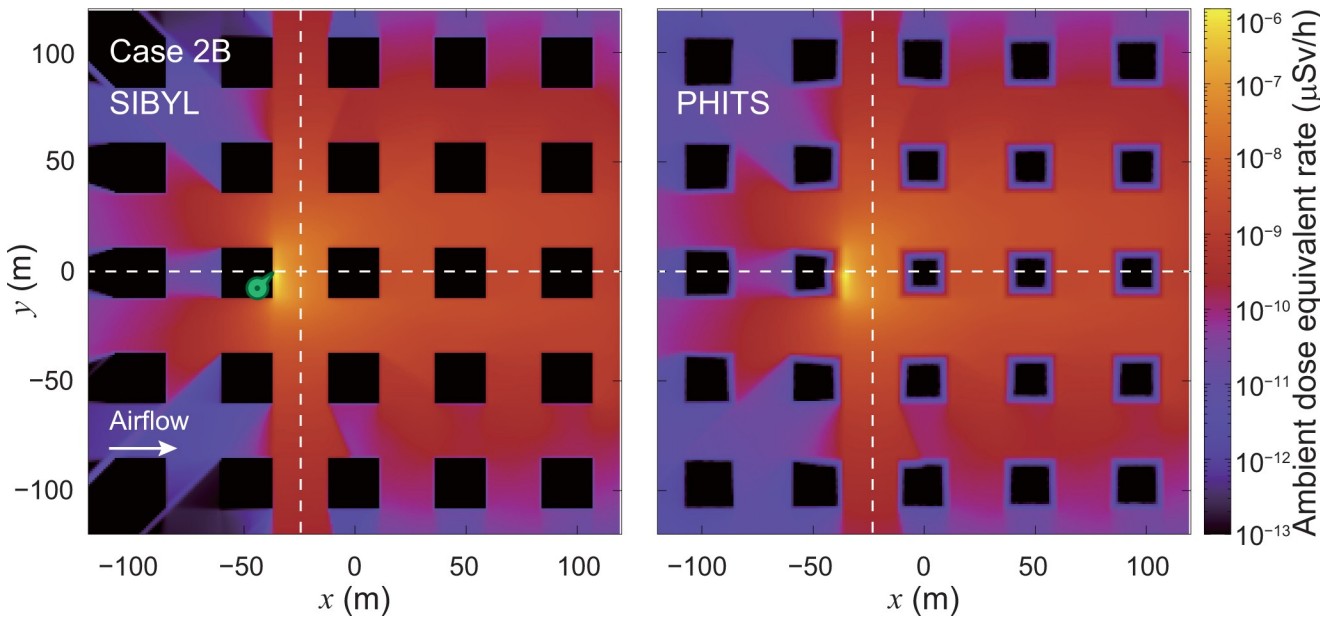

**Fig 17. Distribution map of ambient dose equivalent rate at 1 m above the ground on the grids of the target area for case 2B.** The left and right panels are the calculation results of SIBYL and PHITS, respectively. The dashed lines indicate the lines $y = 0$ m and $x = -25$ m. The marker on the map represents the hypothetical release point at $(-36, 0, 0)$.

inputting and outputting data and resizing the dose-response functions due to the small size of the problem.

Cases 2A and 2B involved dose estimation for unsteady-state dispersion in arrayed buildings considering dose attenuation due to obstacles. In these cases, the dose-attenuation process in the dose-estimation routine dominated the whole computation, with parallel efficiencies reaching 0.94 and 0.89 for cases 2A and 2B, respectively. The computation time of case 2B was approximately one-thirtieth that of case 2A, whereas its problem was approximately one-third the size of that of case 2A. This was because SIBYL skipped dose calculations for cells with zero radioactivity to reduce computation time. In case 2B, radionuclides were deposited on the surface of the geometry. Most of the cells in the atmosphere had zero radioactivity; however, they were counted for estimating the problem size. This is also why the computation times of cases 1A and 2B were at the same level even though case 2B was larger than case 1A.

As an example, **Fig 19** demonstrates the results of parallel performance evaluation for case 2B using four nodes in the computer cluster system. One node contains two processors, each with 12 cores. The dashed line represents optimal performance, and the results close to the line show good performance. The open circles and cross marks show the results of parallel computation with OpenMP and MPI, respectively. The asterisks indicate the results of the hybrid parallel computation based on OpenMP and MPI. In the hybrid computation, the number of processing elements was increased with OpenMP threads up to 12, and MPI was applied over 12 for every set of 12 threads. The numbers in **Fig 19** with the letters T and P refer to the number of OpenMP threads and MPI worker processes, respectively.

With fewer than 24 processing elements within one node, the speed-up factors of each parallel computation show good linearity with the number of processing elements, and the parallel efficiencies were more than 0.89. However, speed-up factors of MPI relative to the optimal performance showed increasing deviation with an increasing number of processing elements. As mentioned in the design of parallel computing algorithm section, the MPI manager process

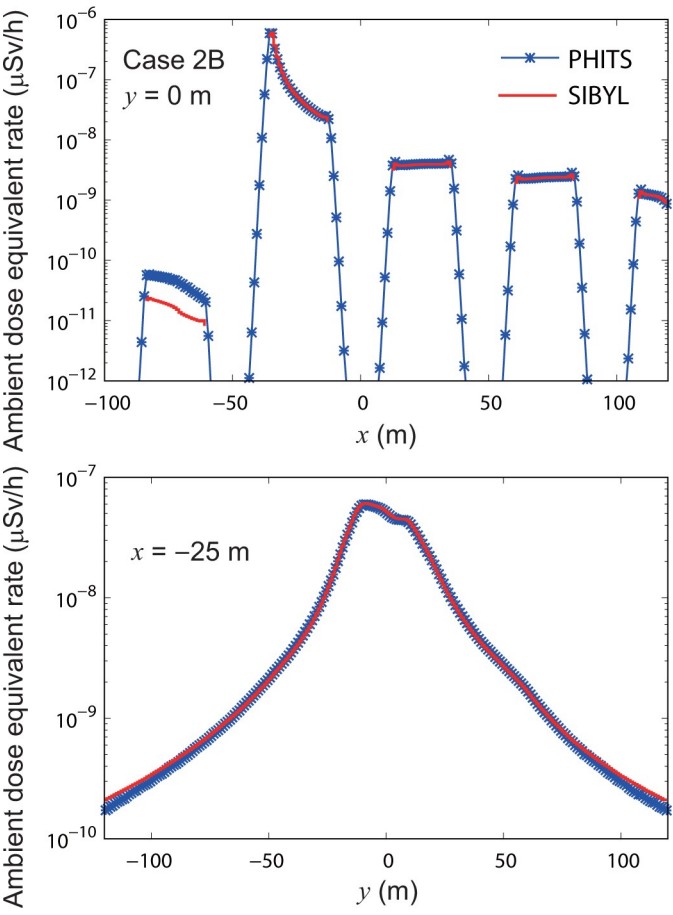

**Fig 18. Distributions of ambient dose equivalent rates on specific lines in the target area for case 2B.** The upper panel shows the calculation results of SIBYL and PHITS on the line $y = 0$ m. The lower panel represents the calculation results on the line $x = -25$ m.

communicates with the worker processes to distribute a task and gather results. The queueing algorithm of this communication is based on a first-in first-out method. The probability of coincidence increased with an increasing number of worker processes, and the parallel efficiencies were degraded to process those data queues. However, the speed-up factors of the

**Table 3. Summary of parallel computing performance by SIBYL for the cases in this study.**

| Case | 1A | 1B | 1C | 2A | 2B |
|---|---|---|---|---|---|
| $N_T \times N_R \times N_{Sz}$ | $3.31 \times 10^{11}$ | $2.70 \times 10^{8}$ | $2.70 \times 10^{8}$ | $4.62 \times 10^{12}$ | $1.44 \times 10^{12}$ |
| $T_{Serial}$ | 4,767.14 | 5.12 | 4.97 | 142,256.18 | 4,607.21 |
| $T_{Parallel}$ | 67.36 | 2.02 | 2.02 | 1580.84 | 54.01 |
| $S$ | 70.77 | 2.53 | 2.46 | 89.99 | 85.30 |
| $\varepsilon$ | 0.74 | 0.03 | 0.03 | 0.94 | 0.89 |

$N_T$ represents the number of cells in the target area and $N_R$ represents the number of cells in the resized dose-response function in a horizontal plane. $N_{Sz}$ is the number of vertical cells in the source region. $T_{Serial}$ and $T_{Parallel}$ indicate the elapsed times in seconds for serial and hybrid parallel computation, respectively. Hybrid parallel computation used 96 processing elements with 12 threads × 8 MPI worker processes. $S$ and $\varepsilon$ are the speed-up factor and parallel efficiency defined in Eq (4), respectively.

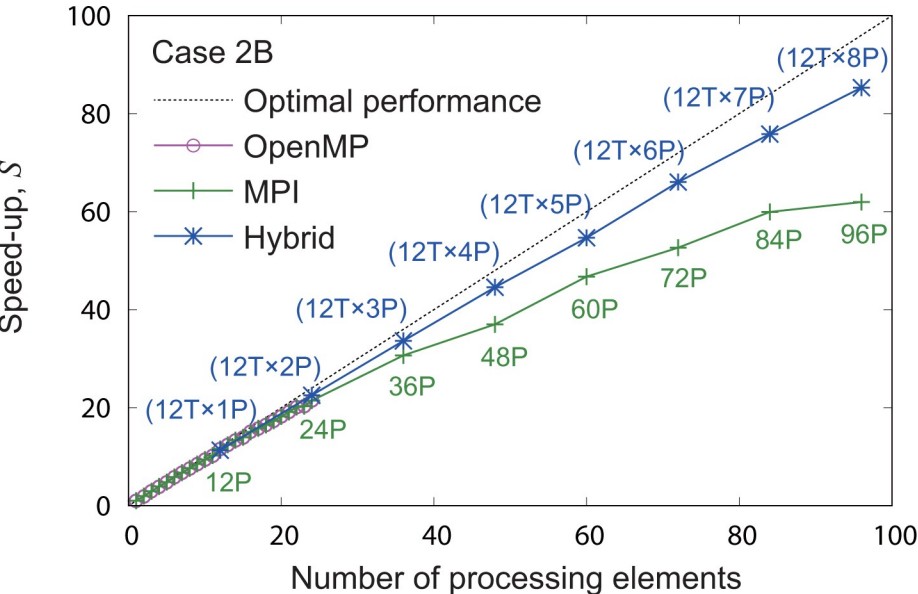

**Fig 19. Performance evaluation of the parallel computation on SIBYL for case 2B.** The numbers followed by the letters T and P represent the number of OpenMP threads and MPI worker processes used in the computation, respectively. The numbers in parentheses indicate the number of combined OpenMP threads and MPI worker processes for the hybrid computation.

hybrid computation maintained linearity to 96 processing elements, that is, 12 threads with 8 MPI worker processes, and the parallel efficiency under this condition was 0.89. The hybrid method could suppress the number of MPI processes using OpenMP threads and avoid the delays related to MPI communication owing to the supervision of the number of MPI processes. Thus, this finding yields good results in terms of parallel efficiency. The tendency of parallel computing performance for the other cases was also same as that shown in **Fig 19**, except for cases 1B and 1C which had computational burdens that were too light to use parallel computation.

In addition, compared with the full Monte Carlo simulations, SIBYL's computation was more than 100 times faster than that of PHITS under the same computational conditions. This was because PHITS traced the vast number of radiation trajectories by considering physical interactions with environmental media in the huge simulation geometry and obtained a good statistical uncertainty for the Monte Carlo simulation. However, SIBYL could avoid those time-consuming simulations by using pre-calculated lattice dose-response functions.

In this research, we selected typical cases for which SIBYL would be used. Consequently, SIBYL with the hybrid parallel computing algorithm achieved the dose calculations within an hour for all cases. Thus, we concluded that SIBYL is applicable for emergency dose assessments as an initial response to a nuclear accident.

## Conclusions

In this research, we developed a simulation code, named SIBYL, for estimating external gamma-ray doses emitted by radionuclides in the atmosphere and on the ground. SIBYL can be coupled with the local-scale atmospheric dispersion model LOHDIM-LES and can estimate ground-level dose rates based on the dispersion and deposition of radionuclides simulated by LOHDIM-LES. To perform quick and accurate dose estimations, SIBYL employed dose-response functions consisting of pre-calculated matrices that included dose contributions

from primary and secondary gamma rays scattered in the air and soil. Moreover, SIBYL was equipped with the unique ability to calculate dose distributions by considering two factors: dose attenuation resulting from obstacles and changes in terrain elevation inside a calculation domain.

The accuracy of SIBYL was examined for steady-state Gaussian plumes over terrain and unsteady-state plumes in building arrays by comparing its results with those of the Monte Carlo radiation-transport code PHITS. The results of SIBYL and PHITS showed good agreement in general, with some discrepancies in the relatively low-dose areas behind obstacles, where radiation transport in the complex geometry played an important role in dose estimation. Moreover, SIBYL showed good performance in hybrid parallel computation using OpenMP and MPI technologies. SIBYL successfully returned results in minutes for the typical cases investigated in this research by using 96 processing elements. PHITS took several hours or days to provide this output under the same computational conditions. From these results, we concluded that SIBYL is applicable for establishing a dose-distribution map inside a target area in less than one hour as an initial response to a nuclear emergency. For a more detailed analysis of the retrospective dose assessments after the emergency phase, the use of full Monte Carlo simulations by PHITS becomes an option with conversion code developed in this research to convert the outputs of LOHDIM-LES to the inputs of PHITS.

The coupled system of LOHDIM-LES and SIBYL is currently undergoing tests for dose estimations around a fuel reprocessing facility in Japan. A performance examination of the coupled simulations under real meteorological and topographical conditions has already been performed for the routine release of $^{85}$Kr from the facility's ventilation shaft, and the simulation results were compared with data measured by radiation monitors. The results agreed well with the measured data, and these results will be presented elsewhere. Furthermore, we will prepare a graphical user interface to control SIBYL calculations and a data viewer based on the open-source multi-platform data analysis and visualization tool ParaView [28]. SIBYL can connect with LOHDIM-LES and other Eulerian-type atmospheric dispersion models simulated on a Cartesian grid system. We will provide the SIBYL code at no cost to the public as open source software.

## Supporting information

**S1 Table. The numerical values plotted in Fig 10.**
(XLSX)

**S2 Table. The numerical values plotted in Fig 12.**
(XLSX)

**S3 Table. The numerical values plotted in Fig 13.**
(XLSX)

**S4 Table. The numerical values plotted in Fig 15.**
(XLSX)

**S5 Table. The numerical values plotted in Fig 16.**
(XLSX)

**S6 Table. The numerical values plotted in Fig 18.**
(XLSX)

**S7 Table. The numerical values plotted in Fig 19.**
(XLSX)

## Acknowledgments

We express our gratitude to Dr. M. Takeyasu of the JAEA for fruitful discussions related to computer codes for emergency responses. We also acknowledge the help provided by the operation team of the Center for Computational Science and E-system (CCSE) at the JAEA. The simulations reported in this study were performed on the computer cluster system maintained by the CCSE.

## Author Contributions

**Conceptualization:** Daiki Satoh, Hiromasa Nakayama.

**Funding acquisition:** Daiki Satoh.

**Investigation:** Daiki Satoh.

**Methodology:** Daiki Satoh, Hiromasa Nakayama, Takuya Furuta.

**Software:** Daiki Satoh, Tamotsu Yoshihiro, Kensaku Sakamoto.

**Writing – original draft:** Daiki Satoh.

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
