## [Decision Letter · Decision Letter 0]

12 Nov 2020

PONE-D-20-30769

Simulation code for estimating external gamma-ray doses from a radioactive plume and contaminated ground using a local-scale atmospheric dispersion model

PLOS ONE

Dear Dr. Satoh,

Thank you for submitting your manuscript to PLOS ONE. After careful consideration, we feel that it has merit but does not fully meet PLOS ONE’s publication criteria as it currently stands. Therefore, we invite you to submit a revised version of the manuscript that addresses the points raised during the review process.

Please, consider important to answer to each one of the suggestions and comments provided by both reviewers. I suggest also, attending to one of the sugestions, putting the manuscript in review by a native english speaking.

We look forward to receiving your revised manuscript.

Kind regards,

Antonio Leal, Ph.D.

Academic Editor

PLOS ONE

Journal Requirements:

[This study was partially supported by the Japan Society for the Promotion of Science (https://www.jsps.go.jp/english/index.html) KAKENHI Grant Number JP17K07017. The funders had no role in study design, data collection and analysis, decision to publish, or preparation of the manuscript.].   

We note that one or more of the authors are employed by a commercial company: Hewlett-Packard Japan, Ltd

Reviewers' comments:

Reviewer's Responses to Questions

**Comments to the Author**

1. Is the manuscript technically sound, and do the data support the conclusions?

Reviewer #1: Yes

Reviewer #2: Yes

2. Has the statistical analysis been performed appropriately and rigorously? 

Reviewer #1: N/A

Reviewer #2: Yes

3. Have the authors made all data underlying the findings in their manuscript fully available?

Reviewer #1: Yes

Reviewer #2: Yes

4. Is the manuscript presented in an intelligible fashion and written in standard English?

Reviewer #1: No

Reviewer #2: Yes

5. Review Comments to the Author

Reviewer #1: In my view the manuscript fulfils the criteria for publication listed in https://journals.plos.org/plosone/s/reviewer-guidelines#loc-criteria-for-publication

with an exception of criterion no.5 regarding "standard English".

Therefore the manuscript merits publication with minor revisions.

The manuscript needs revision in regards to English language. I do not have the time to carry out a language revision of the entire manuscript.

Lines 12-13: “… maps calculated using SIBYL agreed within 10% with those calculated using PHITS…”. I think what is actually meant is that “… maps calculated using SIBYL differ by up to 10% from those calculated using PHITS…”.

Line 409: “The results showed good agreement within 3%.” It should be “The results showed good agreement, with differences up to 3%”.

Line 427, 451-452 and 472: “The results of SIBYL agreed with those of PHITS within x%”. It should be “The results of SIBYL differed from those of PHITS by up to x%”.

Line 496 and 529: “the results of SIBYL agreed within x% with those of PHITS”. It should be “the results of SIBYL differed by up to x% from those of PHITS”.

Other minor comments are listed follow:

Line 40: Replace “they could not” by “they were not designed to”

Lines 90-92: “The reliability of LOHDIM-LES was validated by comparing with the results of the wind tunnel experiments [15, 16] and the field experiments [17] in an actual urban area under real meteorological conditions.” References 15, 16 and 17 describe the experiments. As the sentence is written it gives the impression that they describe the validation of LOHDIM-LES. No references of the validation studies of LOHDIM-LES are actually given. The sentence should be rephrased and references of validation studies must be added (if they exist).

Figures 14 and 17: It would be helpful to show the wind direction on the plots

Reviewer #2: In this study, a system was developed to evaluate ambient dose rates at 1 m height due to gamma-ray source distributed in air and on ground surface utilizing dose-response functions which were prepared beforehand based on accurate Monte Carlo simulations. The system was coupled with atmospheric dispersion models and designed to take source distribution data directly from the dispersion simulations. Furthermore, the system can simulation variation of terrain and shielding effects by objects existing in the environment using simple manners. Ambient dose rates at 1 m height is a basic quantity to indicate the radiation level of environment, and many trials have been made to evaluate them from sources distributed in the environment. The presented system is judged to contribute to improving the dose evaluation in the environment according to several new devices. The manuscript is well-organized; while adding some descriptions is expected to promote a better understanding of readers. The manuscript should be accepted with small modifications.

1. lines 30-31:

It is stated that gamma-ray monitoring cannot cover the whole area. This may not be true; aerial monitoring can cover almost the whole targeted area. Correctly saying, environmental monitoring cannot obtain the whole spatiotemporal information.

2. page 8, 1st paragraph:

It is not mentioned how the depth distribution of deposited radionuclides was assumed. Did you assume that deposited radionuclides distributed just on the flat ground surface? If so, the ambient dose rates may be a little overestimated because ground surface has some roughness and deposited radionuclides may cause initial migration into the ground in the real environment. How do you evaluate these effects?

3. page 8:

It is not clear if electron transport in the environmental media was considered in the simulation using PHITS. It should be described.

4. lines 207-210:

It would be kind to add some explanation on the resizing of dose-response functions. For example, in case of 5 m resolution, do you sum up dose-response functions for every combination (25 x 25 cases) of 1 m square elements included in a source cell and a target cell and average the summed value?

5. page 10, equation (1):

It is not explained how the ranges of summation were determined; that is, how L, M and N were determined. Obviously, they must be large enough to cover the whole source components which contribute to the ambient dose rate; while, they need to be as small as possible to reduce the computational time. How did you determine them? Did you change them according to gamma-ray energy?

5. page 12, equation (3):

It is necessary to explain how the system deal with gamma-ray transport from source deposited on the elevated terrain to the unelevated terrain, and vice versa. Is it not simulated at all?

6. Figures:

In many figures, “ambient dose equivalent” in the y-axis should be “ambient dose equivalent rate”.

7. Results and Discussion:

Difference in computing time between SIBYL and PHITS should be discussed in Results and Discussion. Some descriptions on the computing time difference exist in Abstract and Conclusion, but not in Results and Discussion. Basically, what described in Abstract and Conclusion need to be discussed in the main body of the manuscript.

6. PLOS authors have the option to publish the peer review history of their article (what does this mean?). If published, this will include your full peer review and any attached files.

Reviewer #1: No

Reviewer #2: No

---

## [Author Response · Author response to Decision Letter 0]

4 Dec 2020

5. Review Comments to the Author

Reviewer #1: In my view the manuscript fulfils the criteria for publication listed in https://journals.plos.org/plosone/s/reviewer-guidelines#loc-criteria-for-publication

with an exception of criterion no.5 regarding "standard English".

Therefore the manuscript merits publication with minor revisions.

The manuscript needs revision in regards to English language. I do not have the time to carry out a language revision of the entire manuscript.

Thank you for pointing that out. The re-submitted paper has been corrected grammatically and semantically by an expert editor who speaks native English. The changes regarding English language are highlighted in blue in the marked-up copy of our manuscript.

Lines 12-13: “… maps calculated using SIBYL agreed within 10% with those calculated using PHITS…”. I think what is actually meant is that “… maps calculated using SIBYL differ by up to 10% from those calculated using PHITS…”.

According to the reviewer's suggestion, we revised the sentences.

Line 12-13:

“The dose-distribution maps calculated using SIBYL differed by up to 10% from those calculated using PHITS”

Line 409: “The results showed good agreement within 3%.” It should be “The results showed good agreement, with differences up to 3%”.

According to the reviewer's suggestion, we revised the sentences.

Line 408:

“These results showed good agreement, with differences of up to 3%.”

Line 427, 451-452 and 472: “The results of SIBYL agreed with those of PHITS within x%”. It should be “The results of SIBYL differed from those of PHITS by up to x%”.

According to the reviewer's suggestion, we revised the sentences.

Line 426:

“The results of SIBYL differed from those of PHITS by up to 5%”

Line 449-450:

“The results of SIBYL differed from those of PHITS by up to 5%”

Line 469-470:

“The results of SIBYL differed from those of PHITS by up to 10%”

Line 496 and 529: “the results of SIBYL agreed within x% with those of PHITS”. It should be “the results of SIBYL differed by up to x% from those of PHITS”.

According to the reviewer's suggestion, we revised the sentences.

Line 492-493:

“the results of SIBYL differed from those of PHITS calculated with the ideal radiation absorber by up to 1%”

Line 526:

“the results of SIBYL differed by up to 3% from those of PHITS.”

Other minor comments are listed follow:

Line 40: Replace “they could not” by “they were not designed to”

According to the reviewer's suggestion, we revised the sentences.

Line 38-40:

“they were not designed to assess ground-shine doses from contaminants that were non-uniformly deposited on the ground after the plume passed.”

Lines 90-92: “The reliability of LOHDIM-LES was validated by comparing with the results of the wind tunnel experiments [15, 16] and the field experiments [17] in an actual urban area under real meteorological conditions.” References 15, 16 and 17 describe the experiments. As the sentence is written it gives the impression that they describe the validation of LOHDIM-LES. No references of the validation studies of LOHDIM-LES are actually given. The sentence should be rephrased and references of validation studies must be added (if they exist).

Thank you for pointing that out to be clear the ambiguity of the references. As you pointed out, the Refs. [15-17] are the papers regarding the experiments. But, in the series of the references for LOHDIM-LES [7-11], the calculation results of LOHDIM-LES were compared with the results of those experiments. We clearly state that by revising the sentences as follows:

Line 88-90:

“Previous studies [7–11] validated the reliability of LOHDIM-LES by comparing it with the results of wind tunnel experiments [15, 16] and field experiments [17] in an actual urban area under real meteorological conditions.”

Figures 14 and 17: It would be helpful to show the wind direction on the plots

According to the reviewer's comment, we updated Figures 14 and 17.

Reviewer #2: In this study, a system was developed to evaluate ambient dose rates at 1 m height due to gamma-ray source distributed in air and on ground surface utilizing dose-response functions which were prepared beforehand based on accurate Monte Carlo simulations. The system was coupled with atmospheric dispersion models and designed to take source distribution data directly from the dispersion simulations. Furthermore, the system can simulation variation of terrain and shielding effects by objects existing in the environment using simple manners. Ambient dose rates at 1 m height is a basic quantity to indicate the radiation level of environment, and many trials have been made to evaluate them from sources distributed in the environment. The presented system is judged to contribute to improving the dose evaluation in the environment according to several new devices. The manuscript is well-organized; while adding some descriptions is expected to promote a better understanding of readers. The manuscript should be accepted with small modifications.

1. lines 30-31:

It is stated that gamma-ray monitoring cannot cover the whole area. This may not be true; aerial monitoring can cover almost the whole targeted area. Correctly saying, environmental monitoring cannot obtain the whole spatiotemporal information.

According to the reviewer's comment, we revised the sentences.

Line 29-31:

“they cannot obtain complete spatiotemporal information in locations, especially urban areas, where radioactive materials have been intentionally or accidentally distributed.”

2. page 8, 1st paragraph:

It is not mentioned how the depth distribution of deposited radionuclides was assumed. Did you assume that deposited radionuclides distributed just on the flat ground surface? If so, the ambient dose rates may be a little overestimated because ground surface has some roughness and deposited radionuclides may cause initial migration into the ground in the real environment. How do you evaluate these effects?

Thank you for the valuable comment. Actually, the radionuclides were located just on the flat ground. We did not consider the ground roughness and initial migration. To state that clearly, we revised the sentences as follows.

Line 169-173:

“Ground sources distributed uniformly within the 1 × 1 m2 area with radioactivity of 1 kBq were placed on the ground surface inside a square domain from −500.5 m to 500.5 m centered on the target receptor. We assumed the condition of just after radionuclide descent from the atmosphere onto the ground and did not consider ground roughness and initial migration into the ground.”

3. page 8:

It is not clear if electron transport in the environmental media was considered in the simulation using PHITS. It should be described.

According to the reviewer's comment, we added the following sentences.

Line 163-165:

“Although PHITS simulated the interactions and transport for gamma rays and electrons, only gamma rays including bremsstrahlung were scored at the target receptor to convert gamma-ray fluences to doses using the conversion coefficients.”

4. lines 207-210:

It would be kind to add some explanation on the resizing of dose-response functions. For example, in case of 5 m resolution, do you sum up dose-response functions for every combination (25 x 25 cases) of 1 m square elements included in a source cell and a target cell and average the summed value?

Yes, exactly. According to the reviewer's comment, we revised the sentences as follows.

Line 209-212:

“The numerical values of the resized dose-response functions are computed using an arithmetic mean of data from corresponding elements of the original dose-response functions. Additionally, the values along the z axis are obtained by logarithmic interpolation based on the original data at specific altitudes.”

5. page 10, equation (1):

It is not explained how the ranges of summation were determined; that is, how L, M and N were determined. Obviously, they must be large enough to cover the whole source components which contribute to the ambient dose rate; while, they need to be as small as possible to reduce the computational time. How did you determine them? Did you change them according to gamma-ray energy?

Thank you for providing the important comment. The size of the response functions is fixed to 1001 m × 1001 m in horizontal plane, which is a square domain from -500.5 m to 500.5 m depicted in Fig. 2. The size is enough to cover the whole source components which contribute to the dose at one target cell. According to the reviewer's comment, we revised the sentences. 

Line 177-179:

“A previous study [24] reported that the size of the dose-response function was sufficient to consider the dose contributions from distant sources of the radionuclides.”

Line 207-208:

“The spatial resolution of the original dose-response functions is 1 m inside a 1001 m × 1001 m horizontal domain (as described above),”

Line 218-221:

“The region of the dose-response functions is divided into 2L + 1, 2M + 1, and N number of cells with respective resized resolutions in the l and m horizontal axes and n vertical axis on the Cartesian coordinate system of those response functions (Fig 4). The region of the dose-response functions is truncated if there are division remainders.”

5. page 12, equation (3):

It is necessary to explain how the system deal with gamma-ray transport from source deposited on the elevated terrain to the unelevated terrain, and vice versa. Is it not simulated at all?

SIBYL was not designed to treat the gamma rays from sources deposited on the elevated terrain. The function was prepared for calculations of cloud shine from overhead plume. According to the reviewer's comment, we revised and added the sentences.

Line 253-256:

“This function was designed to estimate cloud-shine doses from an overhead radioactive plume, and thus is not applicable to ground-shine dose calculations from contaminated ground. In addition, radiation shielding by elevated soil was not considered even in the cloud-shine calculations.”

6. Figures:

In many figures, “ambient dose equivalent” in the y-axis should be “ambient dose equivalent rate”.

According to the reviewer's comment, we revised Figures 9–18.

7. Results and Discussion:

Difference in computing time between SIBYL and PHITS should be discussed in Results and Discussion. Some descriptions on the computing time difference exist in Abstract and Conclusion, but not in Results and Discussion. Basically, what described in Abstract and Conclusion need to be discussed in the main body of the manuscript.

According to the reviewer's comment, we revised the sentences and changed the paragraph structure.

Line 612-625:

“The tendency of parallel computing performance for the other cases was also same as that shown in Fig 19, except for cases 1B and 1C which had computational burdens that were too light to use parallel computation.

In addition, compared with the full Monte Carlo simulations, SIBYL’s computation was more than 100 times faster than that of PHITS under the same computational conditions. This was because PHITS traced the vast number of radiation trajectories by considering physical interactions with environmental media in the huge simulation geometry and obtained a good statistical uncertainty for the Monte Carlo simulation. However, SIBYL could avoid those time-consuming simulations by using pre-calculated lattice dose-response functions.

 In this research, we selected typical cases for which SIBYL would be used. Consequently, SIBYL with the hybrid parallel computing algorithm achieved the dose calculations within an hour for all cases. Thus, we concluded that SIBYL is applicable for emergency dose assessments as an initial response to a nuclear accident.”

Line 645-646:

“PHITS took several hours or days to provide this output under the same computational conditions.”

---

## [Decision Letter · Decision Letter 1]

11 Jan 2021

Simulation code for estimating external gamma-ray doses from a radioactive plume and contaminated ground using a local-scale atmospheric dispersion model

PONE-D-20-30769R1

Dear Dr. Satoh,

We’re pleased to inform you that your manuscript has been judged scientifically suitable for publication and will be formally accepted for publication once it meets all outstanding technical requirements.

Kind regards,

Antonio Leal, Ph.D.

Academic Editor

PLOS ONE

Additional Editor Comments (optional):

Reviewers' comments:

Reviewer's Responses to Questions

**Comments to the Author**

1. If the authors have adequately addressed your comments raised in a previous round of review and you feel that this manuscript is now acceptable for publication, you may indicate that here to bypass the “Comments to the Author” section, enter your conflict of interest statement in the “Confidential to Editor” section, and submit your "Accept" recommendation.

Reviewer #1: All comments have been addressed

Reviewer #2: All comments have been addressed

2. Is the manuscript technically sound, and do the data support the conclusions?

Reviewer #1: (No Response)

Reviewer #2: Yes

3. Has the statistical analysis been performed appropriately and rigorously? 

Reviewer #1: (No Response)

Reviewer #2: Yes

4. Have the authors made all data underlying the findings in their manuscript fully available?

Reviewer #1: (No Response)

Reviewer #2: Yes

5. Is the manuscript presented in an intelligible fashion and written in standard English?

Reviewer #1: (No Response)

Reviewer #2: Yes

6. Review Comments to the Author

Reviewer #1: (No Response)

Reviewer #2: (No Response)

7. PLOS authors have the option to publish the peer review history of their article (what does this mean?). If published, this will include your full peer review and any attached files.

Reviewer #1: No

Reviewer #2: No

---

## [Editor Report · Acceptance letter]

13 Jan 2021

PONE-D-20-30769R1 

Simulation code for estimating external gamma-ray doses from a radioactive plume and contaminated ground using a local-scale atmospheric dispersion model 

Dear Dr. Satoh:

I'm pleased to inform you that your manuscript has been deemed suitable for publication in PLOS ONE. Congratulations! Your manuscript is now with our production department. 

Kind regards, 

on behalf of

Professor Antonio Leal 

Academic Editor

PLOS ONE